# GraphFM: A generalist graph transformer that learns transferable representations across diverse domains

**Divyansha Lachi**                                     *div11@upenn.edu*
*University of Pennsylvania*
*Philadelphia, Pennsylvania*

**Mehdi Azabou**                                        *ma4766@columbia.edu*
*Columbia University*
*New York, New York*

**Vinam Arora**                                         *vinam@upenn.edu*
*University of Pennsylvania*
*Philadelphia, Pennsylvania*

**Eva L. Dyer**                                         *dyer1@upenn.edu*
*University of Pennsylvania*
*Philadelphia, Pennsylvania*

**Reviewed on OpenReview:** *https://openreview.net/forum?id=sZTpRfRUtR*

## Abstract

Graph neural networks (GNNs) are often trained on individual datasets, requiring specialized models and significant hyperparameter tuning due to the unique structures and features of each dataset. This approach limits the scalability and generalizability of GNNs, as models must be tailored for each specific graph type. To address these challenges, we introduce GRAPHFM, a scalable multi-graph pretraining approach designed for learning across diverse graph datasets. GRAPHFM uses a Perceiver-based encoder with learned latent tokens to compress domain-specific features into a shared latent space, enabling generalization across graph domains. We propose new techniques for scaling up graph training on datasets of different sizes, allowing us to train GRAPHFM on 152 distinct graph datasets, containing a total of 7.4 million nodes and 189 million edges. This allows us to study the effect of scale on pretraining across domains such as molecules, citation networks, and product graphs, and show that training on diverse datasets improves performance over single-source pretraining. Additionally, pretraining with a mixture of synthetic and real graphs enhances adaptability and stability, leading to competitive performance with state-of-the-art models across various node classification tasks. This approach reduces the burden of dataset-specific training and provides a single generalist model capable of performing across multiple diverse graph structures and tasks. Code is available at `https://github.com/nerdslab/GraphFM`.

## 1 Introduction

Graphs are a fundamental data structure in biology, social systems, and recommendation platforms (Hamilton et al., 2017b). However, many graph neural network (GNN) architectures are specialized for particular regimes and struggle to generalize across others (Topping et al.; Yan et al., 2022; Zhu et al., 2020). Methods that perform well on homophilic graphs, such as citation networks, often degrade on heterophilic graphs that are common in social or biological domains (Abu-El-Haija et al., 2019; Yan et al., 2022). This specialization fragments model development and limits scalability, since each dataset class often requires architecture redesign and extensive hyperparameter tuning (Galkin et al.; Wang et al., 2025; Zhao et al., 2024b). A

generalist approach that performs robustly across diverse graph structures without per-dataset customization is therefore highly desirable.

A central challenge is to integrate graphs that differ in topology, features, and size while still enabling transfer of useful inductive biases across domains. Without a shared representation space or vocabulary for graph structure, learned patterns do not transfer reliably (Galkin et al.). At the same time, experience from language and vision suggests that scaling model capacity and data diversity can unlock transferable capabilities (Wei et al.; Kaplan et al., 2020). Realizing similar benefits for graphs requires architectures that can process large and heterogeneous inputs efficiently while preserving domain-invariant structure (Xia & Huang, 2024).

We introduce GRAPHFM, a multi-graph pretraining framework that addresses these needs. GRAPHFM uses a Perceiver-style encoder (Jaegle et al., 2021; Azabou et al., 2023a) in which a fixed set of learnable latent tokens attends to the input node sequence through cross-attention. The latent tokens act as virtual nodes that compress each graph into a compact representation, which decouples compute from graph size and creates a shared latent space across datasets. This design provides a common interface for heterogeneous graphs and supports efficient scaling.

To train our generalist model, we curate a corpus of 152 datasets for pretraining, including 80 real-world graphs from multiple domains and 72 synthetic graphs that enrich underrepresented regimes such as low homophily. The full corpus contains more than 7.4 million nodes and 163.9 million edges. We exclude popular benchmark datasets from pretraining in order to evaluate generalization on unseen graphs.

Our experiments reveal that scaling both model capacity and data diversity produces consistent and measurable gains in generalization on graphs unseen during training. As we increase model size from 389K to 75M parameters and the number of pretraining tokens from 200K to 7.3M, performance on unseen datasets improves monotonically, with the largest model achieving a 2.1% increase in accuracy over smaller configurations. Training on diverse domains further enhances this effect: adding biological graphs improves accuracy even on citation datasets, and including synthetic graphs leads to the strongest overall results. Beyond accuracy, GRAPHFM exhibits strong generalist behavior with minimal adaptation. Using a fixed learning rate and weight decay, our lightweight MLP fine-tuning (MFT) approach achieves rapid convergence within 10 to 20 steps and strong out-of-the-box performance, while node-decoder fine-tuning (NFT) achieves competitive performance with other state-of-the-art graph transformers. Finally, our sensitivity analysis shows that GRAPHFM maintains stable performance across a wide range of hyperparameters, unlike GCN and NAGphormer, which are highly sensitive to parameter settings. Together, these results demonstrate that a single pretrained model can provide robust, transferable representations across heterogeneous graphs without per-dataset tuning.

The main contributions of this work are as follows:

- **Scalable Pre-training Approach:** We introduce a scalable framework for pretraining on diverse graphs using a Perceiver-based encoder with latent tokens, which efficiently handles graphs with varying sizes and topologies. Our approach includes advanced multi-graph sampling techniques that optimize GPU utilization, enabling large-scale pretraining across a wide range of graph datasets.

- **Demonstration of Benefits from Across-Graph Pretraining:** We show that pretraining on diverse graphs significantly improves the model's ability to generalize and transfer knowledge to unseen graphs. To further enrich diversity, we incorporate synthetic graphs that capture underrepresented structures such as low-homophily patterns and show that this can help improve performance. This demonstrates that a generalist model can leverage common structural features across different datasets to outperform specialized models.

- **Scaling Analysis and Impact of Multi-Graph Pretraining:** We provide the first scaling analysis for multi-graph pretraining on different domains, showing that larger models pretrained on more diverse graph datasets result in better generalization.

## 2 Background

In this section, we provide background on graph transformers, focusing on tokenization of graphs and positional encodings.

### 2.1 Transformers and Self-Attention

Transformers model sequential data by operating on a set of tokens $\mathbf{X} = [\mathbf{x}_1, \ldots, \mathbf{x}_N]$, where dependencies are captured through self-attention (Vaswani, 2017):

$$\text{Attn}(\mathbf{Q}, \mathbf{K}, \mathbf{V}) = \text{softmax}\left(\frac{\mathbf{Q}\mathbf{K}^\top}{\sqrt{d_k}}\right)\mathbf{V},$$

with $\mathbf{Q}, \mathbf{K}, \mathbf{V}$ denoting learned projections of the input tokens. Extending this framework to graphs requires a tokenization scheme that maps graph-structured inputs into a sequence of tokens, while incorporating structural information derived from the adjacency matrix.

### 2.2 Tokenization of Graphs

Given a graph $\mathcal{G} = (\mathcal{V}, \mathcal{E})$ with node features $\{\mathbf{u}_i\}_{i=1}^{|\mathcal{V}|}$, each node $v_i \in \mathcal{V}$ is represented by a token embedding that concatenates a projection of its raw features with a positional encoding:

$$\tilde{\mathbf{u}}_i = \text{MLP}(\mathbf{u}_i) \in \mathbb{R}^{d_f}, \qquad \mathbf{x}_i = (\tilde{\mathbf{u}}_i; \mathbf{p}_i) \in \mathbb{R}^{d_f + d_p},$$

where MLP denotes a multi-layer perceptron, $(\cdot; \cdot)$ is concatenation, and $\mathbf{p}_i \in \mathbb{R}^{d_p}$ encodes structural information from the graph topology. The complete graph is then expressed as a sequence of tokens,

$$\mathbf{X} = [\mathbf{x}_1, \mathbf{x}_2, \ldots, \mathbf{x}_{|\mathcal{V}|}].$$

Since graphs lack a canonical node ordering, the tokenization scheme must be permutation-invariant, ensuring that reindexing nodes does not alter the resulting sequence.

### 2.3 Positional Encodings and Sign-Invariance

To incorporate structural information, each token is assigned a positional embedding derived from the graph topology. A standard approach is to use the eigenvectors of the normalized Laplacian $\mathcal{L} = I - D^{-1/2}AD^{-1/2}$, where $A$ is the adjacency matrix and $D$ the degree matrix. Let $\mathbf{v}_1, \ldots, \mathbf{v}_k$ denote the first $k$ eigenvectors of $\mathcal{L}$, corresponding to the smallest eigenvalues. For node $i$, we construct a vector

$$\mathbf{s}_i = [\mathbf{v}_{1i}, \mathbf{v}_{2i}, \ldots, \mathbf{v}_{ki}] \in \mathbb{R}^k,$$

where $\mathbf{v}_{ji}$ is the $i$-th entry of eigenvector $\mathbf{v}_j$. These eigenvector values provide a continuous notion of position that captures global graph structure and are invariant to permutations of node indices.

A key limitation of spectral encodings is the non-uniqueness of Laplacian eigenvectors: each eigenvector can be multiplied by $-1$ without affecting validity, and for repeated eigenvalues, any orthonormal basis of the eigenspace may be chosen. As shown in Lim et al., these ambiguities make raw eigenvectors unsuitable as features, since they can lead to unstable and inconsistent encodings across graphs.

To address this, SignNet introduces functions that are provably invariant to both global sign flips and basis changes (Lim et al.), while retaining the ability to approximate common positional encodings such as heat kernels and random walks. This property is particularly important in our multi-graph pretraining setting, where consistent positional encodings are required to align structural information across different graphs. We pass $\mathbf{s}_i$ through SignNet to obtain the final positional embedding $\mathbf{p}_i \in R^{d_p}$. This transformation ensures that the resulting positional encoding are both sign- and basis-invariant, while preserving global structural information across graphs.

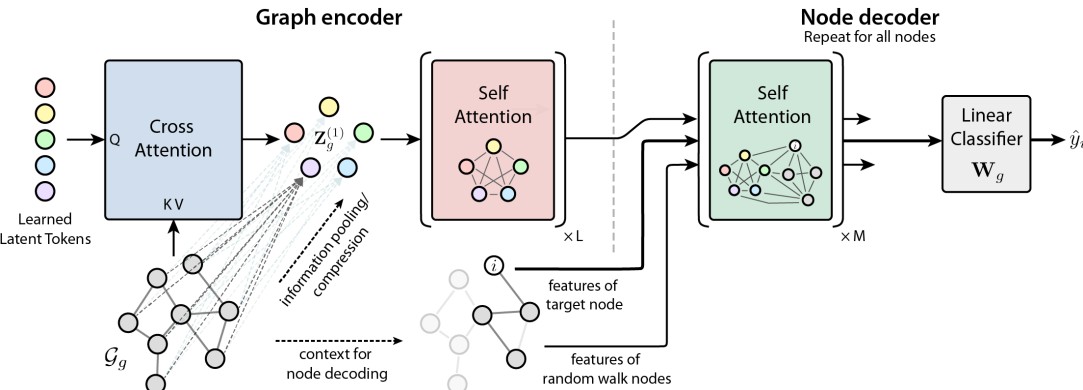

Figure 1: **Overview of GraphFM architecture and multi-graph training approach**: The input node-level tokens are passed through a cross-attention layer, followed by multiple self-attention layers to generate a compressed graph-level representation (latents). We decode node-level properties by creating a spatial sequence with features from a query node, a subset of its neighbors (sampled via a random walk; see Appendix C.2) and the latents, which is then processed by a node decoder that uses self attention across the sequence.

## 3 Methods

In this section, we describe our method, including the model architecture and tokenization (Section 3.1.1 ), our proposed multi-task node decoder for jointly solving node classification and regression tasks by querying from the latent space (Section 3.1.2 ), and efficient tools for scaling (Section 3.2) that allowed us to build a large pretrained model that could integrate the extreme diversity in our pretraining set.

### 3.1 Model

#### 3.1.1 Tokenizing Diverse Graphs with a Perceiver Encoder

Each graph is represented as a sequence of tokens, as described in Section 2. Formally, let $\mathcal{D} = \{\mathcal{G}_g\}_{g=1}^G$ denote a dataset of $G$ graphs, where each graph $\mathcal{G}_g = (\mathcal{V}_g, \mathcal{E}_g)$ has node features $\{\mathbf{u}_i\}_{i=1}^{N_g}$, with $N_g = |\mathcal{V}_g|$. For each graph, we construct a token sequence $\mathbf{X}_g = [\mathbf{x}_1, \ldots, \mathbf{x}_{N_g}]$, where each token $\mathbf{x}_i$ combines a projection of the node features with a positional encoding (see Section 2).

To enable training across diverse graphs with heterogeneous node features and sizes, we map all graphs into a shared latent space using a Perceiver encoder (Jaegle et al.). The encoder learns a fixed set of latent query tokens that attend to the input graph via cross-attention, producing a compact latent representation. In the context of graphs, this can be viewed as routing communication between distant nodes through a small number of learnable virtual nodes that summarize the input graph (Figure 1).

Specifically, we maintain a shared sequence of $K$ learned latent tokens $\mathbf{Z}_0 = [\mathbf{z}_{0,1}, \ldots, \mathbf{z}_{0,K}]$, where $\mathbf{z}_{0,i} \in \mathbb{R}^D$ and $K \ll N_g$ (we use $K = 512$ in this work). Each graph's node embeddings are compressed into this latent space through a cross-attention operation:

$$\mathbf{Z}_g^{(1)} = \text{Cross-Attn}(\mathbf{Q}, \mathbf{K}_g, \mathbf{V}_g) = \mathbf{Z}_0 + \text{softmax}\left(\frac{\mathbf{Q}\mathbf{K}_g^\top}{\sqrt{d_k}}\right)\mathbf{V}_g, \tag{1}$$

where the queries $\mathbf{Q} = \mathbf{W}_q\mathbf{Z}_0$ are projections of the learnable latent tokens, and the keys and values, $\mathbf{K}_g = \mathbf{W}_k\mathbf{X}_g$ and $\mathbf{V}_g = \mathbf{W}_v\mathbf{X}_g$, are projections of the graph's token embeddings. The projection matrices $(\mathbf{W}_q, \mathbf{W}_k, \mathbf{W}_v)$ are shared across all graphs. This is followed by $L$ self-attention blocks operating in the latent space, yielding the final sequence of $K$ latent tokens, $\mathbf{Z}_g^{\text{out}}$. Each block uses standard transformer components with pre-normalization and feed-forward layers (Vaswani, 2017). The overall complexity is $O(KN_g + LK^2)$, which is significantly lower than the $O(N_g^2)$ cost of full self-attention when $K \ll N_g$.

Compressing each graph into a fixed set of latent virtual nodes enables the model to learn a shared vocabulary across graphs and domains, capturing recurring semantic and topological motifs. Moreover, because the bulk

of computation occurs in the latent space, this design naturally accommodates graphs of variable sizes while maintaining constant computational cost per graph.

### 3.1.2 Node decoder

Our encoder model is designed to do the bulk of the computation when processing the graph. To be able to readout node-level features, we developed a multi-task node decoder that combines the virtual node embeddings learned by our encoder $\mathbf{Z}_g^{\text{out}}$ with local information from a node and its neighbors to create a sequence $\mathbf{S}_g^i$ that can be processed by a transformer to produce a final node-level estimate of its class information.

The sequence $\mathbf{S}_g^i$ for the $i^{\text{th}}$ node can be represented as:

$$\mathbf{S}_g^i = \Big[ \underbrace{(\mathbf{x}_i; \tau_{\text{self}})}_{\text{node}}, \underbrace{(\mathbf{x}_{\mathcal{N}_i^1}; \tau_{\text{neighbor}}) \ldots (\mathbf{x}_{\mathcal{N}_i^T}; \tau_{\text{neighbor}})}_{\text{neighbors}}, \underbrace{(\mathbf{Z}_1^{\text{out}}; \tau_{\text{latent}}) \ldots (\mathbf{Z}_K^{\text{out}}; \tau_{\text{latent}})}_{\text{virtual latent nodes}} \Big], \tag{2}$$

where $\mathbf{x}$ and $\tau_{\text{type}}$ denote the features and their token type (latent, self, or neighbor), respectively, and $\mathcal{N}_i^j$ denotes the $j^{\text{th}}$ neighbor selected in the neighborhood of node $i$. We use a small encoder-only transformer with a depth of $M$ to obtain a final set of embeddings $\mathbf{S}_g^{i,\text{out}}$ for node $i$. Note that the complexity is $N_g M (K + T + 1)^2 \ll N_g^2$.

### 3.1.3 Multi-task pretraining on a variety of node classification and regression tasks

In the end, a per-dataset linear classifier (or regressor) $\mathbf{W}_g^T$ is tasked with producing the final predictions $\hat{y}_i$ for node $i$, mapping the final embedding of node $i$, the first token in the $\mathbf{S}_g^i$ sequence in Eq. 2, to the output space as:

$$\hat{y}_i = \mathbf{W}_g^T \mathbf{S}_g^{i,\text{out}}.$$

The linear projection effectively translates the node-level embeddings into task-specific outputs, such as class labels for classification or continuous values for regression. The model handles a wide variety of tasks across different datasets, e.g., citation graphs are trained to predict academic fields, and co-purchasing graphs are used to predict product categories. Each dataset has an arbitrary label space, varying not only in the number of labels but also in the nature and semantics of the output classes.

Since the model is trained end-to-end, it learns how to optimally route and query information on graphs to maximize the performance on the various pre-training tasks. The virtual nodes allow for longer-range and global interactions to be encoded in the virtual node embeddings, and uses this information along with the local information provided by the node's neighbors.

## 3.2 Important ingredients for training on diverse graphs

### 3.2.1 Multi-graph packing

Typically when creating batches for training graph transformers, padding is used to extend the smaller graphs to have the same size as the largest graph in the batch (Rampášek et al., 2022; Ying et al., 2021). This approach is likely inherited from the transformer architectures found in other domains where the context window (or sequence length) is usually fixed. But for graphs, the problem with padding is particularly pronounced when there is a significant size disparity among different graphs in the same batch. Alternative solutions exist, and in particular, the graph community have been pioneers in batching variable-sized graphs. Message-passing frameworks combine multiple graphs into a single large graph over which message passing is conducted (Fey & Lenssen, 2019; Krell et al., 2022). However, these out-of-the-box implementations are not suited for transformers which use fully-connected attention.

To address this, we simply batch graphs by concatenating their node tokens into a single sequence, and use Flash Attention (Dao, 2023) to efficiently handle these variable-length sequences. This eliminates superfluous padding and leads to improved computational efficiency during training.

### 3.2.2 Balanced GPU utilization with the DistributedSSSampler

During multi-GPU distributed training, a global batch is formed by randomly sampling graphs from different datasets, which is then equally split among the GPUs. Naively splitting the batch can lead to unbalanced GPU utilization. On one hand, we can have a large batch of relatively small graphs, and another where we can only have a batch with one or two very large graphs. This means that we would be forced to lower the batch size, to avoid going out of memory when multiple large graphs are batched together. Our *Distributed Snake Strategy Sampler* (DistributedSSSampler) employs a bidirectional filling strategy, where graphs, sorted by their size, are distributed in a snake-like pattern, initially assigned to GPUs from right to left, then left to right and so on. This method effectively pairs large graphs with small ones in subsequent passes, preventing the concentration of multiple large graphs on the same GPU, thus achieving efficient load balancing and uniform GPU utilization. A detailed algorithm and more details are provided in Appendix C.1.

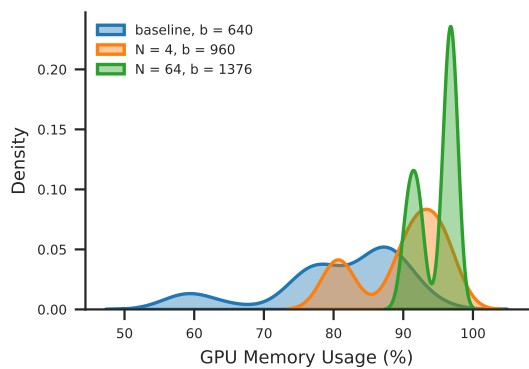

Figure 2: **The computational benefits of using our multi-graph sampling approach**: GPU memory utilization during distributed training using the default batch sampler with 8 GPUs (left), compared to our DistributedSSSampler with $N = 4$ (middle, 4 GPUs and 1 gradient accumulation step) and $N = 64$ (right, 8 GPUs and 8 gradient accumulation steps). The total batch size is $N \times b$.

We show the effectiveness of this approach in Figure 2, where we demonstrate significantly lower variance in GPU load compared to the default PyTorch batch sampler and near 100% utilization. The effectiveness is more pronounced the more GPUs are used[1]. This subsequently allows us to use substantially larger batch sizes, resulting in further improvement in stability and a significant 2-4x speed-up in training time.

### 3.2.3 Overall time and memory savings

In total, our largest model, trained on all the pretraining data, takes ~6 days to train on 8 A40 GPUs for 300 epochs. With our distributed sampler, each epoch takes approximately 56 minutes (0.93 hours), compared to 299.04 minutes (~5 hours) with the standard distributed sampler. By using the distributed sampler, we observe a speedup of approximately 5.53x, reducing the total training time from 33 days to 6 days. Please refer to Appendix C for an ablation study on the proposed sampler and multi-graph packing methods.

## 4 Datasets

In standard practice, one would train on individual datasets, one at a time. However, to build our large multigraph model, we needed to curate a large dataset of graphs that have varied structures, features, and tasks.

**Datasets used for pretraining.** For pre-training, we curated a large set of 80 real-world graph datasets from the PyTorch Geometric library (Fey & Lenssen, 2019) and Network Repository (Rossi & Ahmed) (Figure 3). These datasets span a wide range of domains, including: citation networks, product recommendation graphs, webpage traffic graphs, biological protein-protein interactions, and molecular graphs, and vary in their degree of heterophily (extent to which neighbors share the same class or node-level labels). Each dataset contributes unique structural patterns and tasks, providing a rich source for our model to learn diverse graph representations. In addition to these real-world datasets, we generated 72 synthetic graphs (Tsitsulin et al., 2022) that vary in their hetero- and homophily ratios, and overall size and density (see Appendix B.1). We note that most datasets used in popular benchmarks were left out of pretraining to enable meaningful evaluation of generalization performance on new unseen test datasets

---

[1]The same effect can be obtained using gradient accumulation when resource bound. See Appendix C.1

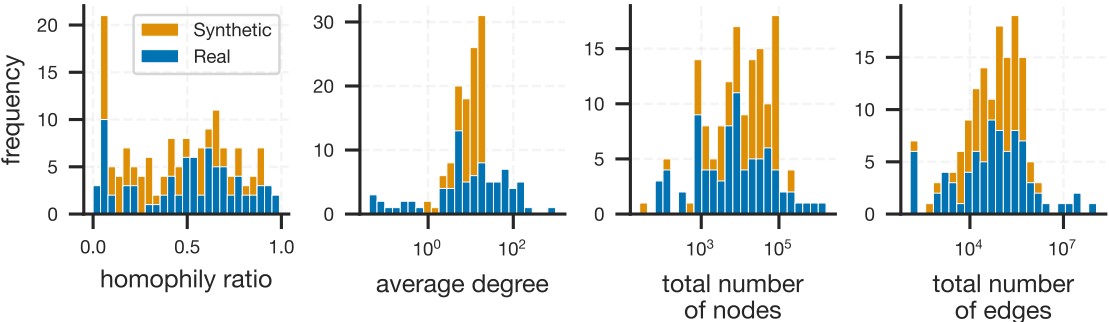

Figure 3: **Characteristics of graph datasets used to train GraphFM:** From left to right, we compute the histograms of the homophily ratio, average degree, number of nodes and number of edges of all 152 graphs used during training. The homophily ratio provides a measure of how frequently a node is directly connected to other nodes from the same class.

In Figure 3, we show a summary of various graph statistics, including the number of nodes and edges, the average degree of each node, and the homophily ratio of the graph. The homophily ratio ranges from 0 to 1 and encodes the average amount of nodes with nearest neighbors from the same class. When comparing our real-world datasets with the synthetic graphs added to the mix (Figure 3), we see a good amount of overlap between most features except for the average degree. The average degree of realworld graphs spans a larger range, and the synthetic graphs have a more limited range. We also find an enrichment of heterophilic graphs with low homophily ratio in the added synthetic data. In total, we counted more than 7.4M nodes and 163.9M edges across all 152 datasets used for pretraining. We point the reader to Appendix B.1 for a detailed description of all datasets.

**Datasets for evaluating generalization to unseen graphs**   To evaluate how general and transferable the learned representations are, we evaluate the model on a set of unseen graph datasets that were excluded during training (see Appendix B.4). These 10 datasets include academic collaboration networks such as "Coauthor-CS" and "Coauthor-Physics" (Sinha et al., 2015) as well as webpage link datasets like "Chameleon" and "Squirrel" (Rozemberczki et al., 2021). The latter are particularly challenging due to their low homophily ratios, where nodes are less likely to connect to others of the same class.

The unseen datasets were not included during training but may share structural similarities with the training data. The label and feature space of these graphs are entirely new to the model, making them suitable for testing generalization. Evaluating on such unseen datasets allows us to examine whether the learned representations can effectively generalize to new graphs with similar structural properties.

## 5  Results

### 5.1  Experimental Setup

**Training:**   To train all of our models, we employed the LAMB optimizer (You et al.) with a learning rate of $10^{-4}$. The learning rate is scheduled based on a linear warmup of 2 epochs, followed by cosine decay until the end of training. We use `bfloat16` mixed-precision and Flash Attention (Dao, 2023) for higher compute efficiency while training. We trained our largest model (75M parameters) for 300 epochs with a batch size of 320 (∼6.4 days) on 8 NVIDIA A40 GPUs. We point the reader to further details on the architecture and model training in Appendix A.1.

**Baselines:**   We compared GraphFM against six baseline models that were consistently reported in both heterophilic and homophilic benchmarks. This included two GNN-based models: GCN (Kipf & Welling, 2016) and GAT (Velickovic et al., 2017); two transformer-based models: SAN (Kreuzer et al., 2021) and NAGphormer (Chen et al.); and two heterophily-based models: MLP and H2GCN (Zhu et al., 2020). For all of the baseline models, we include the best reported accuracy, and when there are no reported results for a dataset we extensively tuned each model as in standard practice (see Appendix B.4). We also provide additional baselines in Appendix D.5 reported for subsets of the datasets tested.

**Evaluation:** To evaluate the quality of the learned representations, we fine-tuned the model on datasets that were excluded from pre-training. We employed two fine-tuning strategies for this purpose. The first strategy, **low-resource MLP fine-tuning (MFT)**, involves freezing both the encoder and node decoder weights, allowing updates only to the feature MLP. This approach evaluates near out-of-the-box performance by leveraging the pretrained model's representations with minimal additional training. The second strategy, **combined MLP and node decoder fine-tuning (NFT)**, provides more flexibility by adapting both the feature MLP and the pretrained node decoder weights, enabling the model to better align with the unseen graph data that it's finetuned on.

For all fine-tuning experiments, we fixed the learning rate to $10^{-3}$ and the weight decay to $10^{-5}$ across all datasets, optimizing with the AdamW optimizer (Loshchilov & Hutter). In our NFT experiments, we additionally applied a gradual unfreezing strategy to update the node decoder weights. Further details are provided in Appendix A.3.

## 5.2 Experiments

### Q1: Is it possible to build a large model spanning many domains?

Recent efforts in graph neural networks (GNNs) have shown success in training models on many graphs (Beaini et al., 2024; Mao et al., 2024). However, these approaches primarily focus on graphs with homogeneous structures, limiting their ability to generalize across different types of graphs. In this experiment, we aim to address a more ambitious question: can we effectively train a large model on diverse, multi-graph datasets that vary significantly in their topologies, features, and downstream classification tasks? Our goal is to determine whether a generalist model can span multiple graph domains and improve performance on new unseen datasets through diverse pretraining.

We trained three different model sizes: a small model with 389K parameters, a medium model with 18M parameters, and a large model with 75M parameters. Each model was pretrained on progressively larger datasets containing different amounts of graph data, ranging from 200K tokens (small), to 2M tokens (medium), and finally to 7.3M tokens (large), created by taking random subsets of the largest dataset (refer to Appendix B.2 for more details). The datasets span a variety of real-world graph types and structures, as described in Section 4. For the largest scale of data, we also introduced synthetic graphs into the mix to further test the model's ability to generalize across highly diverse graph structures. The synthetic graphs provided additional variability in both topologies and node features, allowing us to assess how well the model can handle graph data that extends beyond typical real-world scenarios.

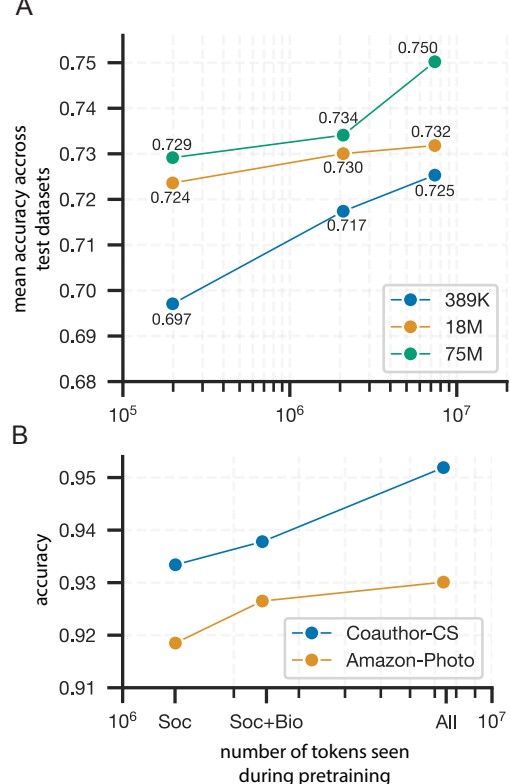

Figure 4: **Scaling analysis showing how increasing model and data scale impacts downstream performance**: **(A)** Average accuracy across test datasets (MFT) for model sizes (389K, 18M, 75M) and token counts (200K, 2M, 7.3M) seen during pre-training, using random splits of the pre-training data. **(B)** Accuracy (MFT) on Coauthor-CS (citation domain) and Amazon-Photo (co-purchasing network) for the 75M model across different domain-wise pre-training splits.

To evaluate how well the pretrained models generalize to new, unseen data, we applied our lightweight MLP fine-tuning approach (MFT) on a set of nine held-out datasets. These include four homophilic datasets (Coauthor-CS, Coauthor-Physics, Amazon-Photo, and Amazon-Comp) and five heterophilic datasets (Texas, Wisconsin, Actor, Squirrel, and Chameleon). As illustrated in Figure 4A, we observe that performance on unseen test datasets improves consistently as the data size increases. Notably,

the largest model, trained on the full 7.3M tokens, achieves a 2.1% improvement in accuracy compared to the smaller models.

We further stratified our pretraining dataset to investigate the effects of cross-domain training by creating three models: (i) "Soc" with social domain graphs (1.3M tokens), (ii) "Soc + Bio" with social and biological graphs (2M tokens), and (iii) "All" with all data, including synthetic graphs (7.3M tokens). As shown in Figure 4B, adding biological datasets improved performance on both Coauthor-CS (citation domain) and Amazon-Photo (co-purchasing network). This suggests that performance continues to scale even if the additional data is from seemingly unrelated domains (refer to Appendix D.1 for additional results from removing synthetic graphs during pretraining).

We want to note that the domains used in this analysis were chosen largely for practical reasons, since they provided the largest sets of available datasets for comparison. As such, they should not be viewed as definitive boundaries, but rather as a convenient stratification to explore the effects of cross-domain pretraining.

These results underscore the importance of both model scale and data diversity. With more data diversity and larger models, the pretrained model demonstrates stronger generalization capabilities. This scaling analysis provides strong evidence that cross-domain pretraining enables better performance, further validating the benefits of training on diverse datasets. Detailed configurations for each model size are provided in Appendix A.1.

**Q2: How does our generalist approach compare with others?**

Next we wanted to understand how our generalist model compares to specialized single-graph approaches. To do this, we evaluated the performance of our largest model (75M) trained on all of the data on standard node classification benchmarks. We focused on the mean rank across datasets as a key metric, providing insight into the model's consistency across diverse graph types. Unlike specialist models that require extensive hyperparameter tuning for each dataset, in MFT we use a single hyperparameter configuration across all evaluations. In the case of NFT, we only have two hyperparameters to tune corresponding to our unfreezing schedule (refer to Appendix A.3.2).

Figure 5 shows the mean rank of our model compared to several common baselines, including message-passing architectures such as GCN (Kipf & Welling, 2016) and GAT (Velickovic et al., 2017), heterophily-based models such as H2GCN (Zhu et al., 2020), and transformer-based architectures such as SAN (Kreuzer et al., 2021) and NAGphormer (Chen et al.). The NFT fine-tuning strategy achieved the best overall rank, demonstrating its flexibility to adapt to a range of graph structures. In addition, the MFT strategy was the second best method and had even lower variance, indicating stable performance of our pretrained models across datasets with varying characteristics.

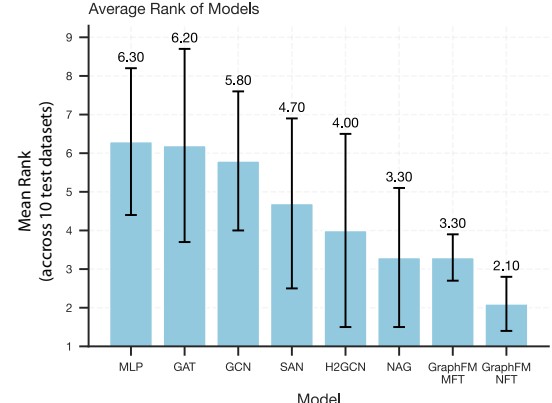

Figure 5: Mean rank of various models across 10 test graph datasets not seen during training (lower is better). Error bars indicate the standard deviation of the ranks.

Specialist models like H2GCN and NAGphormer (NAG) exhibit high variability in their ranking because they perform well on certain datasets but worse on others. H2GCN is designed for heterophilic graphs, while NAG is optimized for homophilic ones. A more detailed comparison to specialist models, including per-dataset performance, can be found in Appendix B.6.

These patterns highlight the trade-offs inherent in models tailored for specific graph types, which may impact their consistency across diverse datasets. In contrast, our single model, using a fixed hyperparameter configuration, maintains a competitive ranking across all datasets without requiring dataset-specific tuning.

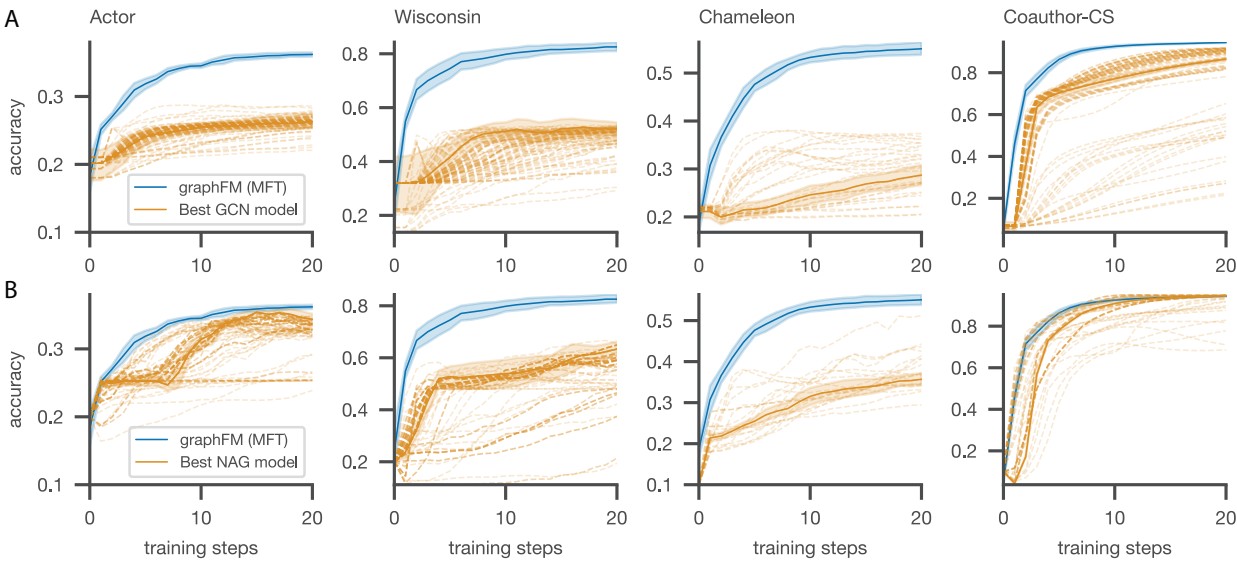

Figure 6: **Analysis of the learning dynamics showing GraphFM achieves faster and more stable convergence compared to baseline single dataset models:** Learning dynamics for 100 (A) random GCN and (B) NAG (NAGphormer) models compared against our lightweight finetuned model GraphFM (MFT) for four datasets. GRAPHFM works out of the box and achieves rapid learning on new datasets with few training steps, while the other approaches are less stable and often require early stopping with decreased performance over training.

### Q3: How does our model generalize out-of-the-box?

A major challenge in applying graph-based models is the extensive tuning often required to achieve competitive performance. Most models are highly sensitive to hyperparameters like learning rate, depth, and weight decay. Tuning these hyperparameters, especially across datasets with different graph topologies and sizes, requires significant time and computational resources, and even then, finding a good configuration can be difficult (Guo et al., 2022; Tsitsulin et al., 2022).

In contrast, GRAPHFM offers strong out-of-the-box performance without requiring any significant tuning. To demonstrate this, we evaluated GRAPHFM using the same fixed learning rate and weight decay across multiple datasets (learning rate $= 10^{-3}$, weight decay $= 10^{-5}$) and observed stable and high performance across all datasets (Figure 6). Fine-tuning GRAPHFM with our simple MFT strategy resulted in low variance and rapid convergence, without the need for extensive hyperparameter exploration. This makes GRAPHFM highly efficient and cost-effective compared to models that require substantial tuning.

To highlight this contrast, we compared the performance of GRAPHFM with 100 randomly configured versions of GCN and the best performing transformer-based NAGphormer (Chen et al.)(Figure 6). Both baseline models exhibit a wide range of performance depending on the hyperparameter choices, with some configurations leading to significant instability or poor results. For instance, in the Texas dataset, GCN required exhaustive exploration of hyperparameter settings to find a stable and effective configura-

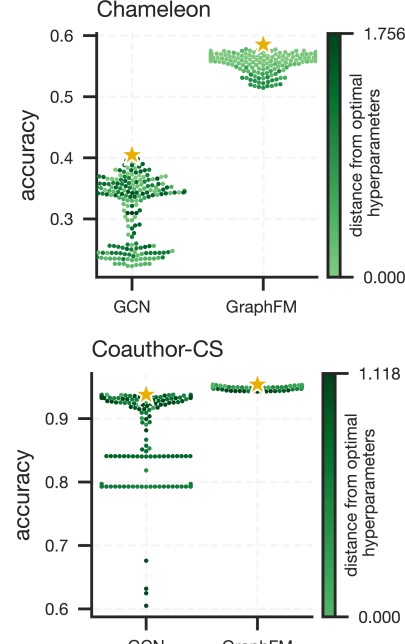

Figure 7: **Comparison of model sensitivity.** The performance of GCN and GRAPHFM for 100 different random hyperparameters on Chameleon and Coauthor-CS. Star denotes the model with the optimal hyperparameters, and the color indicates the $\ell_2$-distance between the optimal solution and each model's hyperparameters.

tion. Similarly, NAGphormer's performance fluctuated greatly depending on the dataset and the selected parameters, further emphasizing the cost of tuning.

Additionally, GRAPHFM demonstrates quick convergence, reaching near-optimal performance within 10-20 training steps (Figure 6), in stark contrast to GCN which required considerably more iterations to converge. This efficiency is a direct result of leveraging a pretrained model, which allows GRAPHFM to start from a robust initialization and quickly adapt to the target task. The reduced need for hyperparameter tuning and faster convergence further solidify the advantages of pretraining in minimizing computational overhead and time-to-solution. Ultimately, our results position GRAPHFM as a cost-effective and reliable choice for a wide range of node classification tasks.

**Q4: How stable are the solutions?**

Graph-based models are highly sensitive to hyperparameter configurations, where even small deviations from optimal settings can lead to substantial performance degradation. This sensitivity poses significant challenges for ensuring stable and robust deployment. Thus, we wanted to examine the stability of model performance by exploring the performance landscape around the optimal hyperparameter configuration. We analyze the performance of both a GCN and GRAPHFM (MFT) on Coauthor-CS (homophilic) and Chameleon (heterophilic) datasets for different hyperparameters around the optimal hyperparameters (Figure 7). This set of hyperparameters is marked with a star, and other models are colored based on the normalized $\ell_2$-distance of their hyperparameter vectors to the optimal hyperparameter vector. For GRAPHFM, we observe that the distribution is concentrated around the optimal point, suggesting low sensitivity to the choice of the hyperparameters used for finetuning. We also observe that the relationship between hyperparameter deviation and performance is linear. On the other hand, for the GCN model, small deviations in hyperparameters can lead to large changes in performance, suggesting instability of the model with respect to the hyperparameters and a much noisier landscape around the optimal model.

# 6 $CO_2$ Emissions

Training our largest model took 1228.8 A40 hrs, totaling an energy usage of 368kWh. The GPUs were housed and powered at an academic datacenter in Georgia, with a carbon energy intensity of 0.35 kgCO$_2$e/kWh and Power Usage Effectiveness of 1.15, the resulting emissions were approximately 148kg of CO$_2$eq.

# 7 Related Work

**Graph foundation modeling approaches.** Foundation models have achieved significant success for language, vision and time-series data (Radford et al., 2018; Dehghani et al., 2023; Das et al., 2024). These models are pre-trained on large datasets and can be adapted to a wide range of downstream tasks, effectively utilizing both prior knowledge from the pre-training stage and data from the downstream tasks to enhance performance (Brown et al., 2020). The concept of foundation models has recently extended into graph learning, leading to the development of Graph Foundation Models (GFMs) (Ibarz et al., 2022; Beaini et al., 2024; Galkin et al.; Mao et al., 2024). These models aim to generalize across diverse graph-structured data by leveraging large-scale pretraining, similar to foundation models in vision and language domains.

Initial GFM efforts primarily focused on domain-specific GFMs, where shared structures and feature vocabularies simplify pretraining. For example, Mole-BERT utilizes pretraining to improve property prediction specifically for molecules and materials (Xia et al.). Additionally, large-scale models like MatterSim (Yang et al., 2024), designed to predict molecular behaviors across different elements and conditions.

Beyond domain-specific applications, Graph Foundation Models (GFMs) are increasingly being developed for general-purpose tasks across diverse graph domains. Similarly, recent advancements have explored scaling laws in graph models, showing that larger models can lead to improved transfer learning and generalization (Liu et al., b), consistent with our work which shows that scale improves performance. Models like Triplet-GMPNN (Ibarz et al., 2022) and ULTRA (Galkin et al.) tackle foundational tasks in algorithmic reasoning and

knowledge graphs respectively, but they are still grounded in narrow domains. Other recent efforts have leveraged LLMs to unify graph inputs via textual representations (Liu et al., a).

Recent work has also begun addressing the more challenging problem of cross-domain generalization, where graphs differ significantly in topology, size, and feature space. For instance, GraphProp (Sun et al., 2025) and GFSE (Chen et al., 2025) focus on structural generalization by pretraining on topological properties such as random walk embeddings. GCOPE (Zhao et al., 2024a) and MDGPT (Yu et al., 2024) incorporate domain-aware components like virtual coordinators or learnable domain tokens to encode domain-specific priors. GraphAny (Zhao et al., 2024b) proposes a fully inductive model that ensembles predictions from analytically derived LinearGNNs. While a lot of progress has been made most existing models are constrained by limited dataset scale—GraphProp and GFSE were trained on 5–8 datasets, and GraphAny only on 1—making it difficult to assess generalization robustness. In contrast, GraphFM is pretrained on 152 graph datasets (7.4M nodes, 189M edges), offering an unprecedented testbed to study scale and diversity effects.

**Scaling up graph transformers.** Graph transformers bypass standard local learning rules in GNNs by allowing all nodes on the graph to interact through self-attention (Dwivedi & Bresson, 2020). However, due to the high computational cost and benefits of the inductive bias in message passing, a number of methods have been proposed to move beyond full self-attention or combine transformers with GNNs. One class of methods combine transformer blocks with GNNs, including GraphTrans (Wu et al., 2021), GraphGPS (Rampášek et al., 2022), and SAT (Chen et al., 2022). Another strategy is to reduce the computational complexity by using the transformer module on a coarsened or compressed graph. For instance, ANS-GT (Zhang et al., 2022) introduced a node-sampling-based graph transformers, incorporating hierarchical attention and graph coarsening, and Gapformer (Liu et al., 2023) uses k-hops local pooling and global pooling to coarsen the large graph into a smaller set of nodes. Exphormer (Shirzad et al., 2023) coarsens the graph by doing computation through expander graphs (Deac et al., 2022). This idea of compression has also been studied through the lens of "skeletonization" (Cao et al., 2024) where they learn to identify uninformative background nodes and use this information to achieve competitive performance with as little as 1% of the nodes in the graph. Many of these approaches leverage virtual nodes to facilitate message passing across large graphs; however, the compression techniques used in these works are often based on heuristics like pooling layers or expander graphs, in contrast to our work where the compression is fully learned through the use of the Perceiver encoder to aggregate information into a set of latent tokens.

## 8   Conclusion

In this paper, we introduced GRAPHFM, a multi-graph pretraining framework for scalable learning across diverse graph datasets. We curated 152 datasets for pretraining, including 80 real-world graphs and 72 synthetic graphs, totaling more than 7.4M nodes and 163.9M edges. Standard benchmark datasets were excluded from pretraining to enable evaluation on unseen graphs. The synthetic corpus enriched low-homophily regimes and complemented the real-world distributions, which had a broader range of average degrees. This mixture provided a varied training signal across topology, feature space, and label structure.

Our scaling study shows clear benefits from both model capacity and data diversity. Across three model sizes (389K, 18M, 75M parameters) and increasing numbers of pretraining tokens (200K, 2M, 7.3M), accuracy on held-out datasets improved monotonically, with the largest model trained on 7.3M tokens achieving a 2.1% gain over smaller configurations. Domain-stratified experiments further indicate that adding biological graphs improves performance on citation and co-purchase datasets, and that including synthetic graphs yields the strongest overall results. These findings support the view that diversity, not only size, is a key driver of transferable representations.

We evaluated two adaptation strategies on unseen datasets. Low-resource MFT freezes the encoder and node decoder, and adjusts only the feature MLP; NFT additionally updates the node decoder with gradual unfreezing. NFT achieved the best mean rank across ten test graphs, while MFT ranked second with lower variance, which indicates stable generalization without dataset-specific tuning. Learning-dynamics analyses show that GRAPHFM reaches near-optimal accuracy within 10 to 20 steps with a single fixed learning rate and weight decay across tasks, while GCN and NAGphormer exhibit broad performance variability and frequent instability under random hyperparameters.

Sensitivity analyses around the optimal hyperparameters reveal a smooth and forgiving landscape for GRAPHFM. Accuracy degrades approximately linearly with distance from the best settings, which suggests robustness to modest deviations and reduces the operational burden of fine-grained tuning. In contrast, GCN exhibits sharp drops for small changes in configuration, which complicates practical deployment. Together, these results position GRAPHFM as a cost-effective and reliable starting point for new graphs, including challenging low-homophily settings.

Looking ahead, we expect further gains from expanding the pretraining corpus to additional structured domains such as meshes, point clouds, and knowledge graphs, and from broadening the task palette beyond node classification to include link prediction, graph-level prediction, and self-supervised objectives. Progress in synthetic graph generation and principled domain balancing should strengthen scaling trends and improve robustness under distribution shift. A unified suite of cross-domain benchmarks, paired with standardized adaptation protocols like MFT and NFT, can help establish repeatable scaling laws for graph learning and accelerate the development of generalist graph foundation models.

## 9 Acknowledgements and Funding Disclosure

Thanks to Shivashriganesh P. Mahato, Keertika Saroj and Yifei Sun for insightful discussions and feedback on this manuscript. This project was supported by NSF CAREER Award RI:2146072, NSF award CIF:RI:2212182 as well as generous gifts from the CIFAR Azrieli Global Scholars Program.

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

# Appendix

## A    Additional Model Details

### A.1    Model Configuration Details

We used pretrained 3 configuration of models—small (398K), medium (18M) and large (75M)—for our analysis. Details of the configuration for each model are given in Table A1. In the first cross-attention layer, we used Flash Attention, whereas for all subsequent attention layers, we used Memory-Efficient Attention. Both implementations were sourced from the xFormers library (Lefaudeux et al., 2022).

Table A1: Architectural details of GraphFM for different parameter sizes used in Section 3.2

| Parameter Count | 75M | 18M | 389K |
|---|---|---|---|
| **Num Latents** ($K$) | 512 | 256 | 32 |
| **Latent Dimension** | 512 | 256 | 32 |
| **Cross-Attention** | | | |
| Heads | 4 | 4 | 4 |
| FFN hidden dim | 2048 | 1024 | 128 |
| **Self-Attention** | | | |
| Depth ($L$) | 12 | 10 | 4 |
| Heads | 8 | 4 | 4 |
| FFN hidden dim | 2048 | 1024 | 128 |
| **Node Decoder** | | | |
| Depth ($M$) | 4 | 4 | 2 |
| Heads | 8 | 4 | 4 |
| FFN hidden dim | 2048 | 1024 | 128 |

### A.2    Rescaling the learning rates for different graph sizes

When training on variable sized graphs, the MLP and linear decoder for each dataset receive updates based on the number of nodes from their respective datasets present in the batch and thus smaller graphs get updated less frequently when compared to large graphs. To mitigate this imbalance, we implemented dataset-specific learning rates for the feature MLP and linear decoders. Since they receive updates less frequently, when they do, we use a larger learning rate to update them. Without this adjustment, the weights of the common Perceiver encoder and node decoder would advance more quickly than those of the dataset-specific components, potentially leading to suboptimal learning for smaller datasets.

### A.3    Fine-Tuning Strategies

In our evaluation of GraphFM's generalization capability, we employed two fine-tuning strategies aimed at adapting the model to the unseen test datasets.

#### A.3.1    Low-resource MLP Fine-tuning (MFT)

This approach is designed to assess how well the pretrained model performs out-of-the-box on different test graphs without extensive training. In MFT, we freeze the pretrained model and only fine-tune a lightweight multi-layer perceptron (MLP) on top of the learned representations. This strategy allows us to quickly adapt the model to new tasks while retaining the majority of the original learned parameters. MFT is particularly useful in low-resource settings, where computational power or time is limited, as it requires minimal additional training while still providing insight into how well the pretrained model generalizes. For all MFT experiments, we used a learning rate of $10^{-3}$ and a weight decay of $10^{-5}$, optimized using the AdamW optimizer (Loshchilov & Hutter).

### A.3.2   MLP and Node Decoder Fine-tuning (NFT)

In contrast to MFT, the NFT strategy involves fine-tuning part of the model and is recommended when sufficient computational resources are available and the goal is to extract the maximum performance from the model. In NFT, we gradually unfreeze the node decoder, enabling the model to more effectively adapt to the new dataset. Specifically, we set a predefined epoch $U$ at which unfreezing begins, starting from the bottom layers of the node decoder. After every $S$ epochs, additional layers are unfrozen in a bottom-up manner, facilitating gradual transition to full finetuning of the model. Concurrently, the learning rate is decayed by a factor of 1.5 each time a new layer is unfrozen, ensuring controlled parameter updates. For all datasets, we tune the hyperparameters $U$ and $S$, with $U$ set to 10, 20, or 30 epochs and $S$ set to 5 or 10 epochs. This gradual unfreezing mitigates training instability, as smaller perturbations are made to higher-level feature representations. As a result, NFT allows for better adaptation, particularly for unseen testing datasets, and is well suited for cases where exploiting the capacity of pretrained models is critical.

## B   Additional Details on Datasets

### B.1   Pretraining datasets

The largest model (75M parameters) was trained on 80 real world and 72 synthetic datasets. The real world datasets and their characteristics are given in Table A3.

The synthetic datasets were created using the GraphWorld (Palowitch et al., 2022) using the Stochastic Block Model (Holland et al., 1983). The generator parameters are listed in Table A2. In the graph generation process, the *node homophily ratio* is varied. The homophily is given by the following formula:

$$\frac{1}{|\mathcal{V}|} \sum_{v \in \mathcal{V}} \frac{|\{(v, w) : w \in \mathcal{N}(v) \wedge y_v = y_w\}|}{|\mathcal{N}(v)|},$$

where $\mathcal{V}$ denotes the set of all nodes in the graph, $\mathcal{N}(v)$ denotes all the neighbors of an arbitrary node $v$, and $y_v$ denotes the class membership of the node $v \in \mathcal{V}$. We classify datasets into *homophilic datasets* and *heterophilic datasets* based on the homophily score: datasets with homophily $\geq 0.5$ are classified as *homophilic datasets* and *heterophilic datasets* otherwise.

### B.2   Details on small and medium scale dataset

The small and medium scale datasets, as discussed in Section 5.2, were created by taking a random subset of the large dataset(80 real and 72 synthetic).

**Dataset subset for small scale data:**   The following datasets were used to train models with small scale data: Wiki, BlogCatalog, Roman-empire, Minesweeper, Tolokers, Questions, Twitch-EN, Twitch-FR, Twitch-PT, Twitch-RU, DeezerEurope, GitHub, LastFMAsia, Airports-USA, Airports-Europe, PolBlogs and EmailEUCore

**Dataset subset for medium scale data:**   The following datasets were used to train models with medium scale data: Wiki, BlogCatalog, Roman-empire, Minesweeper, Tolokers, Questions, Twitch-EN, Twitch-FR, Twitch-PT, Twitch-RU, DeezerEurope, GitHub, LastFMAsia, Airports-USA, Airports-Europe, PolBlogs and EmailEUCore, Reddit, Reddit2, Flickr, Yelp, PPI, Facebook, Amazon-ratings, Minesweeper, Twitch-DE, Twitch-ES, FacebookPagePage, Airports-Brazil, penn94, reed98, amherst41, johnshopkins55, genius, CitationFull-CiteSeer, CitationFull-Cora-ML and CitationFull-PubMed

### B.3   Details on social and biology domain datasets

The social and biology datasets, as discussed in Section D.1 and Section 5.2, included the following subsets:

Table A2: Graphworld generator parameters for synthetic graphs

| Parameter Name | Description | Values |
|---|---|---|
| nvertex | Number of vertices in the graph. | [32, 500000] |
| $p/q$ ratio | The ratio of in-cluster edge probability to out-cluster edge probability. | [0.1, 10.0] |
| avg. degree | The average expected degrees of the nodes. | [1.0, 20.0] |
| feature center distance | The variance of feature cluster centers, generated from a multivariate Normal. | [0.0, 5.0] |
| num clusters | The number of unique node labels. | [2, 6] |
| cluster size slope | The slope of cluster sizes when index-ordered by size. | [0.0, 0.5] |
| power exponent | The value of the power law exponent used to generate expected node degrees. | [0.5, 1.0] |

**Dataset subset for social domain:** The following datasets were used to train the social-specific model: fb-CMU-Carnegie49, Yelp,Wiki, BlogCatalog, Facebook, Twitch-DE, Twitch-EN, Twitch-ES, Twitch-FR, Twitch-PT, Twitch-RU, DeezerEurope, GitHub, FacebookPagePage, LastFMAsia, penn94, reed98, amherst41, johnshopkins55, genius and soc-pokec.

**Dataset subset for biology domain:** The following datasets were added as part of the biology domain to train the combined social and biology model: BZR, DD, DD199, DD21, DD242, DD244, DD349, DD497, DD6, DD68, DD687, DHFR, ENZYMES, ENZYMES118, ENZYMES123, ENZYMES295, ENZYMES296, ENZYMES297, ENZYMES8, KKI, OHSU, PROTEINS-full, Peking-1, Tox21_p53, gene, proteins-all and PPI.

## B.4 Finetuning Datasets

For our evaluations, we held out a number of datasets that are used for standard benchmarks in both larger scale node classification and heterophilic graphs.

### B.4.1 Homophilic Datasets

We use five real-world datasets, Amazon Computers and Amazon Photos (McAuley et al., 2015), Coauthor CS and Coauthor Physics (Sinha et al., 2015) and Obgn-Arxiv (Hu et al., 2020). Key statistics for the different datasets are listed in Table A3 in the finetuning-section. The experimental setup follows that of (Luo et al., 2022), where we split the dataset into development and test sets. All the hyperparameter tuning is done on the development set and the best models are evaluated on the test set. The runs are averaged over 20 random splits to minimize noise. We follow a 60:20:20% train/val/test split for the Amazon and Coauthor datasets. For Obgn-Arxiv we follow the experimental setup used in (Hu et al., 2020). The results for the Coauthor-Physics, Coauthor-CS, and Amazon-Photos obtained from in Table A8 have been sourced from (Liu et al., 2023). The results for the Amazon-Comp dataset are taken from (Hoang et al., 2023) except for MLP which was obtained from (Luo et al., 2022).

Table A3: Pre-Training Datasets and their characteristics

| Dataset | Number of Graphs | Nodes | Edges | Homophily Ratio | Excess Homophily | Average Degree | Node Features | Node Classes | Learning Rate |
|---|---|---|---|---|---|---|---|---|---|
| BA-1_10_60-L5 | 1 | 804 | 46410 | 0.2 | 0.0004 | 115.45 | 1 | 5 | 0.0014 |
| BA-2_24_60-L2 | 1 | 10693 | 639750 | 0.5 | 0.0014 | 119.66 | 1 | 2 | 0.0087 |
| BZR | 405 | 35.75 | 76.71 | 0.42 | 0.0192 | 0.07 | 1 | 53 | 0.0082 |
| CL-100K-1d8-L9 | 1 | 92482 | 373989 | 0.11 | 0.0005 | 8.09 | 1 | 9 | 0.00064 |
| CL-10K-1d8-L5 | 1 | 10000 | 44896 | 0.2 | 0.0037 | 8.98 | 1 | 5 | 0.00096 |
| DD | 1178 | 284.32 | 1431.32 | 0.07 | 0.0015 | 0.058 | 1 | 89 | 0.00085 |
| DD199 | 1 | 841 | 1902 | 0.067 | 0.0144 | 4.52 | 1 | 20 | 0.00085 |
| DD21 | 1 | 5748 | 14267 | 0.07 | 0.0103 | 4.96 | 1 | 40 | 0.00085 |
| DD242 | 1 | 1284 | 3303 | 0.08 | 0.0265 | 5.14 | 1 | 20 | 0.00042 |
| DD244 | 1 | 291 | 822 | 0.074 | 0.0180 | 5.65 | 1 | 20 | 0.00085 |
| DD349 | 1 | 897 | 2087 | 0.05 | 0.0067 | 4.65 | 1 | 20 | 0.00085 |
| DD497 | 1 | 903 | 2453 | 0.06 | 0.0068 | 5.43 | 1 | 20 | 0.0028 |
| DD6 | 1 | 4152 | 10320 | 0.07 | 0.0126 | 4.97 | 1 | 20 | 0.00085 |
| DD68 | 1 | 775 | 2093 | 0.072 | 0.0048 | 5.4 | 1 | 20 | 0.0028 |
| DD687 | 1 | 725 | 2600 | 0.06 | 0.0050 | 7.17 | 1 | 20 | 0.0028 |
| DHFR | 756 | 42.43 | 89.09 | 0.32 | 0.0189 | 0.04 | 3 | 53 | 0.0018 |
| ENZYMES | 600 | 32.63 | 124.27 | 0.67 | 0.1768 | 0.09 | 18 | 3 | 0.0020 |
| ENZYMES118 | 1 | 96 | 121 | 0.58 | 0.4375 | 2.52 | 1 | 2 | 0.00087 |
| ENZYMES123 | 1 | 90 | 127 | 0.52 | 0.7111 | 2.82 | 1 | 2 | 0.0076 |
| ENZYMES295 | 1 | 124 | 139 | 0.71 | 0.8387 | 2.24 | 1 | 2 | 0.0076 |
| ENZYMES296 | 1 | 126 | 141 | 0.72 | 0.8095 | 2.24 | 1 | 2 | 0.00087 |
| ENZYMES297 | 1 | 122 | 149 | 0.65 | 0.8360 | 2.44 | 1 | 2 | 0.0020 |
| ENZYMES8 | 1 | 88 | 133 | 0.77 | 0.8181 | 3.02 | 1 | 2 | 0.0076 |
| ER-AvgDeg10-100K-L2 | 1 | 99997 | 499332 | 0.50 | 0.0019 | 9.99 | 2 | 2 | 0.0049 |
| ER-AvgDeg10-100K-L5 | 1 | 99997 | 499332 | 0.20 | 0.0014 | 9.99 | 1 | 5 | 0.0013 |
| KKI | 83 | 26.96 | 96.84 | 0 | 0.0 | 0.39 | 1 | 189 | 0.0012 |
| MSRC-21 | 563 | 77.52 | 396.65 | 0.74 | 0.0968 | 0.13 | 1 | 24 | 0.0063 |
| MSRC-21C | 209 | 40.28 | 193.20 | 0.61 | 0.0581 | 0.27 | 1 | 22 | 0.0017 |
| MSRC-9 | 221 | 40.58 | 193.21 | 0.69 | 0.0881 | 0.26 | 1 | 10 | 0.009 |
| OHSU | 79 | 82.01 | 399.32 | 0 | 0.0002 | 0.56 | 1 | 189 | 0.0095 |
| PLC-40-30-L5 | 1 | 11025 | 437979 | 0.2 | 0.0003 | 79.45 | 1 | 5 | 0.0086 |
| PLC-60-30-L2 | 1 | 117572 | 7045181 | 0.5 | 6.2980 | 119.84 | 1 | 2 | 0.0013 |
| PROTEINS-full | 1113 | 39.06 | 145.63 | 0.97 | 0.1916 | 0.05 | 2 | 8 | 0.0063 |
| Peking-1 | 85 | 39.31 | 154.71 | 0 | - | 0.44 | 1 | 189 [1] | 0.0027 |
| SW-10000-6-0d3-L2 | 1 | 10000 | 30000 | 0.5 | 0.0026 | 6 | 1 | 2 | 0.00096 |
| SW-10000-6-0d3-L5 | 1 | 10000 | 30000 | 0.2 | 0.0012 | 6 | 1 | 5 | 0.0088 |
| SYNTHETIC | 300 | 100 | 392 | 0.18 | 0.0374 | 0.16 | 1 | 8 | 0.0018 |
| TerroristRel | 1 | 881 | 8592 | 0.92 | 0.7433 | 19.51 | 1 | 2 | 0.0033 |
| Tox21_p53 | 1 | 153563 | 314046 | 0.62 | 0.0009 | 4.09 | 1 | 46 | 0.00054 |
| fb-CMU-Carnegie49 | 1 | 6637 | 249967 | 0.5 | 0.0697 | 75.33 | 1 | 3 | 0.0010 |
| gene | 1 | 1103 | 1672 | 0.4 | 0.5557 | 3.03 | 1 | 2 | 0.012 |
| proteins-all | 1 | 43471 | 162088 | 0.66 | 0.3710 | 7.46 | 1 | 3 | 0.00075 |
| reality-call | 1 | 27058 | 51200 | 0.9 | 0.0 | 15 | 1 | 2 | 0.0071 |
| Reddit | 1 | 232965 | 114615892 | 0.76 | 0.6529 | 983.98 | 602 | 41 | 0.0035 |
| Reddit2 | 1 | 232965 | 23213838 | 0.78 | 0.6913 | 199.29 | 602 | 41 | 0.0035 |
| Flickr | 1 | 89250 | 899756 | 0.31 | 0.1340 | 20.16 | 500 | 7 | 0.0051 |
| Yelp | 1 | 716847 | 13954819 | - | - | 38.93 | 300 | 100 [1] | 0.00031 |
| Wiki | 1 | 2405 | 17981 | 0.71 | 0.6053 | 14.95 | 4973 | 17 | 0.0012 |
| BlogCatalog | 1 | 5196 | 17981 | 0.40 | 0.2680 | 132.21 | 8189 | 6 | 0.0099 |
| PPI | 1 | 56944 | 1612348 | 0.63 | - | 56.63 | 50 | 121 [1] | 0.0016 |
| Facebook | 1 | 4039 | 88234 | 0.99 | - | 43.69 | 1283 | 193 [1] | 0.0011 |
| Roman-empire | 1 | 22662 | 65854 | 0.05 | 0.0208 | 5.81 | 300 | 18 | 0.0074 |
| Amazon-ratings | 1 | 24492 | 186100 | 0.38 | 0.1266 | 15.2 | 300 | 5 | 0.00082 |
| Minesweeper | 1 | 10000 | 78804 | 0.68 | 0.0094 | 15.76 | 7 | 2 | 0.0088 |
| Tolokers | 1 | 11758 | 1038000 | 0.59 | 0.1801 | 176.56 | 10 | 2 | 0.0022 |
| Questions | 1 | 48921 | 307080 | 0.84 | 0.0790 | 12.55 | 301 | 2 | 0.0061 |
| Twitch-DE | 1 | 9498 | 315774 | 0.64 | 0.1691 | 66.49 | 128 | 2 | 0.0023 |
| Twitch-EN | 1 | 7126 | 77774 | 0.59 | 0.1711 | 21.82 | 128 | 2 | 0.0010 |
| Twitch-ES | 1 | 4648 | 123412 | 0.59 | 0.1634 | 53.10 | 128 | 2 | 0.0011 |
| Twitch-FR | 1 | 6551 | 231883 | 0.54 | 0.0306 | 70.79 | 128 | 2 | 0.0010 |
| Twitch-PT | 1 | 1912 | 64510 | 0.58 | 0.1333 | 67.47 | 128 | 2 | 0.0012 |
| Twitch-RU | 1 | 4385 | 78993 | 0.63 | 0.0787 | 36.02 | 128 | 2 | 0.0011 |
| DeezerEurope | 1 | 28281 | 185504 | 0.52 | 0.03038 | 13.11 | 128 | 2 | 0.0070 |
| GitHub | 1 | 37700 | 578006 | 0.84 | 0.3778 | 30.66 | 128 | 2 | 0.0065 |
| FacebookPagePage | 1 | 22470 | 342004 | 0.88 | 0.8198 | 30.44 | 128 | 2 | 0.00085 |
| LastFMAsia | 1 | 7624 | 55612 | 0.87 | 0.7656 | 14.59 | 128 | 18 | 0.0092 |
| Airports-Brazil | 1 | 131 | 1074 | 0.46 | 0.1303 | 16.39 | 131 | 4 | 0.0013 |
| Airports-Europe | 1 | 399 | 5995 | 0.40 | 0.1930 | 30.05 | 399 | 4 | 0.0015 |
| Airports-USA | 1 | 1190 | 13599 | 0.69 | 0.2371 | 22.85 | 1190 | 4 | 0.0092 |
| PolBlogs | 1 | 1490 | 19025 | 0.91 | 0.8233 | 25.54 | 1 | 2 | 0.0013 |
| EmailEUCore | 1 | 1005 | 25571 | 0.36 | 0.2354 | 50.89 | 1 | 42 | 0.0032 |
| penn94 | 1 | 41554 | 2724458 | 0.51 | 0.0278 | 131.11 | 4814 | 2 | 0.0064 |
| reed98 | 1 | 962 | 37624 | 0.52 | 0.0213 | 78.22 | 745 | 2 | 0.0032 |
| amherst41 | 1 | 2235 | 181908 | 0.53 | 0.0385 | 162.78 | 1193 | 2 | 0.011 |
| johnshopkins55 | 1 | 5180 | 373172 | 0.55 | 0.0628 | 144.08 | 2406 | 2 | 0.0025 |
| genius | 1 | 421961 | 984979 | 0.62 | 0.08040 | 4.67 | 12 | 2 | 0.00040 |
| CitationFull-CiteSeer | 1 | 4230 | 10674 | 0.95 | 0.9437 | 5.04 | 602 | 6 | 0.0011 |
| CitationFull-Cora-ML | 1 | 2995 | 16316 | 0.78 | 0.7401 | 10.89 | 2879 | 7 | 0.0028 |
| CitationFull-PubMed | 1 | 19717 | 88648 | 0.80 | 0.6641 | 8.99 | 500 | 3 | 0.00087 |
| soc-pokec | 1 | 1632803 | 30622564 | 0.44 | - | 37.51 | 500 | 3 | 0.00019 |

[1] Multi label binary classification.

Table A4: Fine-Tuning Datasets and Their Characteristics

| Dataset | Number of Graphs | Nodes | Edges | Homophily Ratio | Average Degree | Node Features | Node Classes |
|---|---|---|---|---|---|---|---|
| Actor | 1 | 7600 | 30019 | 0.21 | 7.89 | 932 | 5 |
| Amazon-Computers | 1 | 13752 | 4491722 | 0.77 | 71.51 | 767 | 10 |
| Amazon-Photo | 1 | 7650 | 238162 | 0.82 | 62.26 | 745 | 8 |
| Coauthor-CS | 1 | 18333 | 163788 | 0.80 | 17.86 | 6805 | 15 |
| Coauthor-Physics | 1 | 34493 | 495924 | 0.93 | 28.75 | 8415 | 5 |
| Chameleon | 1 | 2277 | 36101 | 0.23 | 31.70 | 2325 | 5 |

### B.4.2 Heterophilic Datasets

We use five real-world datasets with graphs that have a homophily level $\leq 0.30$, Texas, Wisconsin and Actor (Pei et al., 2020) and Chameleon and Squirrel (Rozemberczki et al., 2021). Key statistics for the different datasets are listed in Table A3 in the finetuning-section. We follow the experimental setup in (Pei et al., 2020), and use the same 10 train/val/test splits that are provided. The results for GCN based methods and heterophily based methods in Table A9 have been taken from (Azabou et al., 2023b), and the results for transformer based methods have been taken from (Liu et al., 2023)

### B.5 Standard hyperparameter search grid for baselines

The hyperparameter search space grid used for tuning baselines for Table A6 is detailed in Table A5.

Table A5: Hyperparameter Search Space

| Hyperparameter | Type | Range |
|---|---|---|
| Hidden Dim | Categorical | {16, 32, 64, 128} |
| Depth | Categorical | {1, 2} |
| Dropout | Uniform | [0.0, 0.9] |
| Learning Rate | Log uniform | [5e-5, 5e-1] |
| Weight Decay | Log uniform | [1e-5, 1e-2] |

### B.6 Detailed Comparison with Specialist Models

On both homophilic and heterophilic benchmarks (Table A6), GraphFM performs on par with state-of-the-art specialist models trained from scratch on each dataset. While the best-performing baseline model varies across datasets, GraphFM consistently ranks among the top three: the NFT fine-tuning strategy achieves the highest average rank overall, while MFT is tied for second place with NAG. Furthermore, MFT demonstrates significantly lower variance in rank compared to NAG, whose rankings display a more bimodal distribution across datasets. This indicates that GraphFM provides more stable performance across diverse graph structures.

Specialist models such as H2GCN and NAG show variability in performance due to their design focus. H2GCN, tailored for heterophilic graphs, performs strongly on heterophilic datasets but struggles with homophilic ones. Conversely, NAG, optimized for homophilic graphs, excels in homophilic settings but is less effective on heterophilic datasets. These results highlight the trade-offs inherent in models designed for specific graph types, limiting their generalization capabilities across diverse datasets.

In contrast, GraphFM achieves strong performance across all datasets without requiring extensive hyperparameter tuning, unlike the specialist models that were fine-tuned for each dataset. By using a single

Table A6: **Results on a variety of homophilic and heterophilic node classification benchmarks.** From left to right, we show different message passing and graph transformer architectures, and then GRAPHFM in both the lightweight MLP-only finetuning (MFT) and node decoder finetuning (NFT). The top three numbers are bold, with the highest in bright red fading to black. Models are ranked on all 10 datasets and the average and standard deviation ranking is at the bottom.

| | | GCN | MLP | GAT | H2GCN | SAN | NAG | GraphFM-MFT | GraphFM-NFT |
|---|---|---|---|---|---|---|---|---|---|
| Homophilic | Physics | 95.38±0.20 | 95.12±0.26 | 95.14±0.28 | 96.28±0.13 | **96.83±0.18** | **96.66±0.16** | 96.64±0.17 | **96.77±0.12** |
| | CS | 94.06±0.16 | 92.99±0.51 | 93.61±0.14 | 94.02±0.31 | 94.16±0.36 | **95.00±0.14** | **95.19±0.21** | **95.24±0.18** |
| | Photo | 85.94±1.18 | 88.66±0.85 | 87.13±1.00 | 91.56±0.80 | **94.17±0.65** | **94.64±0.60** | 93.01±1.82 | **94.37±0.35** |
| | Computer | 89.47±0.46 | 84.63 | **90.78±0.13** | 89.33±0.27* | 89.83±0.16 | **91.22±0.14** | 89.95±0.83 | **90.07±0.21** |
| | Ogbn arxiv | **70.40±0.10** | 52.63±0.12 | 67.56±0.12 | 68.29±0.67 | 69.17±0.15 | 68.21±0.02* | **69.96±0.21** | **70.01 ± 0.18** |
| Heterophilic | Texas | 55.14±5.16 | **80.81±3.31** | 52.16±6.63 | **84.86±7.23** | 60.17±6.66 | 68.37±5.27* | **80.81±2.76** | **82.16±3.24** |
| | Wisconsin | 51.76±3.06 | **85.29±3.31** | 49.41±4.09 | **87.65±4.98** | 51.37±2.08 | 68.23±5.99* | 83.13±2.35 | **83.62±3.21** |
| | Actor | 27.32±1.10 | **36.63±0.70** | 27.44±0.89 | 35.70±1.00 | 27.32±1.10 | 34.33±0.94* | **36.29±0.63** | **38.01±1.07** |
| | Chameleon | 38.44±1.92 | 46.21±2.99 | 38.44±1.92 | **60.11±2.15** | 44.32±1.73* | 57.39±0.02* | **58.64±1.24** | **59.12±1.64** |
| | Squirrel | 31.52±0.71 | 28.77±1.56 | 36.77±1.68 | 36.48±1.86 | 30.92±2.14* | **49.93±0.07*** | **42.80±1.54** | **42.98±1.62** |
| | **Avg Rank (Homophilic)** | 5.2 ± 2.6 | 7.6 ± 0.9 | 6.0 ± 2.2 | 5.6 ± 0.9 | 3.4 ± 1.5 | 2.8 ± 2.0 | 3.4 ± 0.9 | 2.0 ± 0.7 |
| | **Avg Rank (Heterophilic)** | 6.6 ± 0.5 | 4.0 ± 2.5 | 6.6 ± 1.7 | 2.4 ± 1.9 | 6.6 ± 0.5 | 4.0 ± 1.7 | 3.2 ± 0.4 | 2.0 ± 0.7 |
| | **Avg Rank (Overall)** | 5.9 ± 1.9 | 5.8 ± 2.6 | 6.3 ± 1.9 | 4.0 ± 2.2 | 5.0 ± 2.0 | 3.4 ± 1.9 | 3.3 ± 0.7 | 2.0 ± 0.7 |

* This result was missing from existing literature and was obtained through extensive hyperparameter tuning.

hyperparameter configuration (learning rate $= 10^{-3}$), GraphFM consistently achieves competitive rankings. Additionally, the NFT fine-tuning strategy provides significant benefits for challenging datasets such as Amazon-Photos and Actor, as allowing parts of the model to remain learnable enables better adaptation to out-of-distribution datasets.

## C Additional Details on Multi-Graph Training

One key aspect of our work is testing scale. Thus, to build a model across large amounts of diverse graph data, we developed a number of approaches for efficient training and multi-GPU usage.

Figure A1 shows an ablation study the epoch time for various GPU optimizations we have proposed in Section 3.2. The epoch time was calculated using the medium-sized model with 18M parameters, as detailed in Appendix A.1.

**Note:** Removing chaining made it impossible to run the largest model (75M parameters) with our available computational resources (8 A40 GPUs). Therefore, we performed the ablation using the medium-sized model. This highlights the significance of our optimization techniques, which enabled us to scale up and run such large models efficiently.

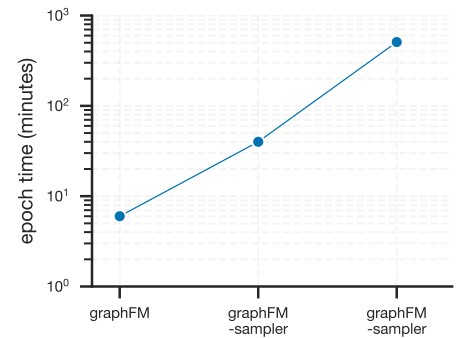

Figure A1: **Ablation for GPU optimizations**: Epoch time in minutes on removing various gpu optimizations proposed for GRAPHFM.

### C.1 DistributedSSSampler

In designing this sampler, we prioritized ensuring that it neither introduces bias into the data sampling process nor alters the distribution of the graphs from the datasets. Its primary function is to enhance batch construction and distribution across GPUs.

First, the sampler defines a set of $N$ buckets with a fixed node budget $B$, where $N$ can be the number of GPUs and $B$ is the node-level batch size. The graphs (across all GPUs) are sorted in descending order based upon their size. The sampler then employs a bidirectional filling strategy within the buckets. The distribution process, as described in Algorithm 1 involves distributing graphs in a snake-like pattern, initially filling from right to left, then switching to left to right and so on. When a graph is added to a bucket, it uses up part of the budget, equal to its size. This method effectively pairs larger graphs with smaller ones in subsequent passes, preventing the concentration of multiple large graphs on the same GPU, thus achieving efficient load balancing and uniform GPU utilization. Figure A2A shows an overview of how the sampler distributes the graphs into buckets. We find that stability is improved with a larger number of buckets $N$ (Figure A2B). When the number of GPUs is fixed, we can achieve a larger $N$ by using gradient accumulation, which artificially increases the number of buckets by a factor equal to the number of accumulation steps, without biasing the sampling process.

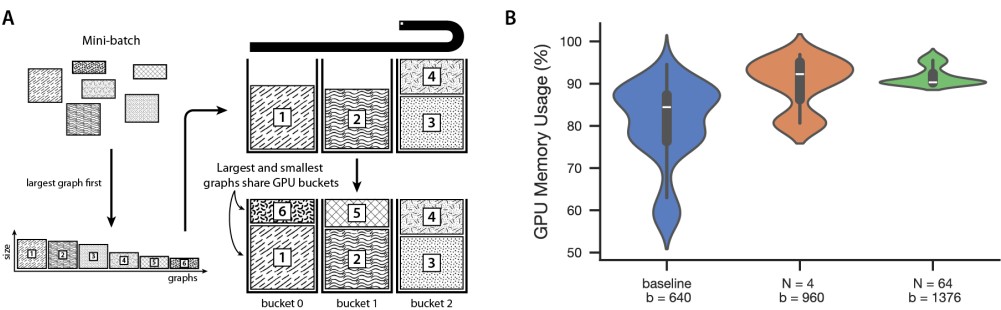

Figure A2: **Multi-GPU utilization**: **A**: A diagram visualizing our sample distribution strategy. **B**: GPU memory utilization during distributed training when using the default batch sampler vs. our DistributedSSSampler for N=4 and N=64 buckets.

---

**Algorithm 1** Distribute graph nodes into virtual GPU buckets

---

1: **input:** Batch size $B$, Bucket count $N$, Graphs in the dataset $\mathcal{G} = \{\mathcal{G}_0, \mathcal{G}_1, \ldots\}$, Subgraphs sampled for this minibatch $\mathcal{G}^m = \{\mathcal{G}_0^m, \mathcal{G}_1^m, \ldots\}$
2: **precondition:** $\sum_i |\mathcal{G}_i^m| = N \times B$
3: **initialize**:
4:     $buckets \leftarrow$ array of $N$ empty arrays                    # will store subgraphs in each bucket
5:     $counts \leftarrow$ array of $N$ zeroes                      # will store number of nodes in each bucket
6:     $b \leftarrow 0$                                      # bucket index
7:     $d \leftarrow 1$                                        # direction
8:     Sort $\mathcal{G}^m$ according to node-counts in $\mathcal{G}$, largest graph goes first
9: **for all** $\mathcal{G}_i^m$ in $\mathcal{G}^m$ **do**
10:     **while** $|\mathcal{G}_i^m| > 0$ **do**
11:       **if** $counts[b] < B$ **then**
12:         # insert a part of $\mathcal{G}_i^m$ into bucket $b$
13:         $n \leftarrow \min(|\mathcal{G}_i^m|,\ B - counts[b])$
14:         $counts[b] \leftarrow counts[b] + n$
15:         append first $n$ nodes of $\mathcal{G}_i^m$ to $buckets[b]$
16:         remove first $n$ nodes from $\mathcal{G}_i^m$
17:       **end if**
18:       # go to the next bucket, switching direction at the boundaries
19:       $b \leftarrow b + d$
20:       **if** $b \geq N$ or $b < 0$ **then**
21:         $d \leftarrow -d$
22:         $b \leftarrow b + d$
23:       **end if**
24:     **end while**
25: **end for**
26: **return** $buckets$

---

## C.2 GraphSAINT Random Walk Sampler

Efficient neighborhood sampling for large graphs is crucial for our node decoder, as traditional methods for k-hop neighborhood sampler often become computationally prohibitive with the increasing size and complexity of the graph data. To overcome these limitations, we have adopted the GraphSAINT Random Walk Sampler (Zeng et al.), specifically designed for efficient sampling in large-scale graphs.

## C.3 RAM Optimization in Multi-GPU Environments

In multi-GPU training environments, efficient use of system memory is crucial, especially when handling large graph datasets. Traditional approaches lead to substantial memory redundancy, as each GPU process typically loads a complete dataset into system RAM. This results in each process duplicating the dataset in system memory, leading to inefficient memory usage and potential system overload.

To address this, we utilize a shared memory management approach using Python's `multiprocessing.Manager()` to coordinate dataset access across multiple GPU processes. This method ensures that each dataset is loaded into RAM only once, regardless of the number of GPUs, thereby avoiding duplication and conserving memory resources.

# D Additional Experiments

## D.1 Separating pretraining datasets into different domains

We further stratified our pretraining dataset to investigate the effects of cross-domain training, and created three models that contained: (i) graph datasets from "social domains" including product graphs and citation networks (1.3M tokens), (ii) both the social datasets and all biological graphs in the dataset (Bio+Soc, 2M tokens), and (iii) compare with our model trained on all data including sytnthetic graphs (7.3M tokens).

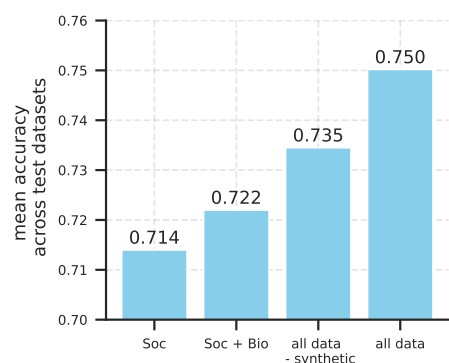

Figure A3: **Domain Scaling**: Average accuracy across unseen testing datasets (using MFT) for models trained on different subsets of data

When comparing graph features across social and biological domains, we found distinct structural differences: biological datasets generally exhibited higher levels of heterophily, lower average degree, and fewer edges, whereas social graphs showed more homophily, higher degrees, and denser connections (Figure A4B). Synthetic graphs added a wide range of characteristics, particularly increasing the number of heterophilic graphs used in pretraining, which contributed to a broader diversity of features (Figure A4A). To isolate the contribution of synthetic graphs, we further stratified the pretraining data to include only real-world datasets.

All three models were then fine-tuned on four homophilic datasets (coauthor-CS, coauthor-physics, amazon-photos, and amazon-computers) and five heterophilic datasets (Texas, Wisconsin, Actor, Squirrel, and Chameleon) held out for fine-tuning.

As shown in Figure A3, incorporating biological datasets, despite being seemingly unrelated to the target domains, improves performance on unseen test datasets, with mean accuracy increasing from 0.714 (Soc) to 0.722 (Soc + Bio). This suggests that knowledge learned from the biology domain positively impacts performance in other domains. Extending pretraining to all available real-world datasets from Table A3 yields additional improvements, with mean accuracy reaching 0.735. Finally, adding synthetic graphs boosted performance even more, with a mean accuracy of 0.750, indicating that diversity (not just domain specific data) is the key to improving generalization.

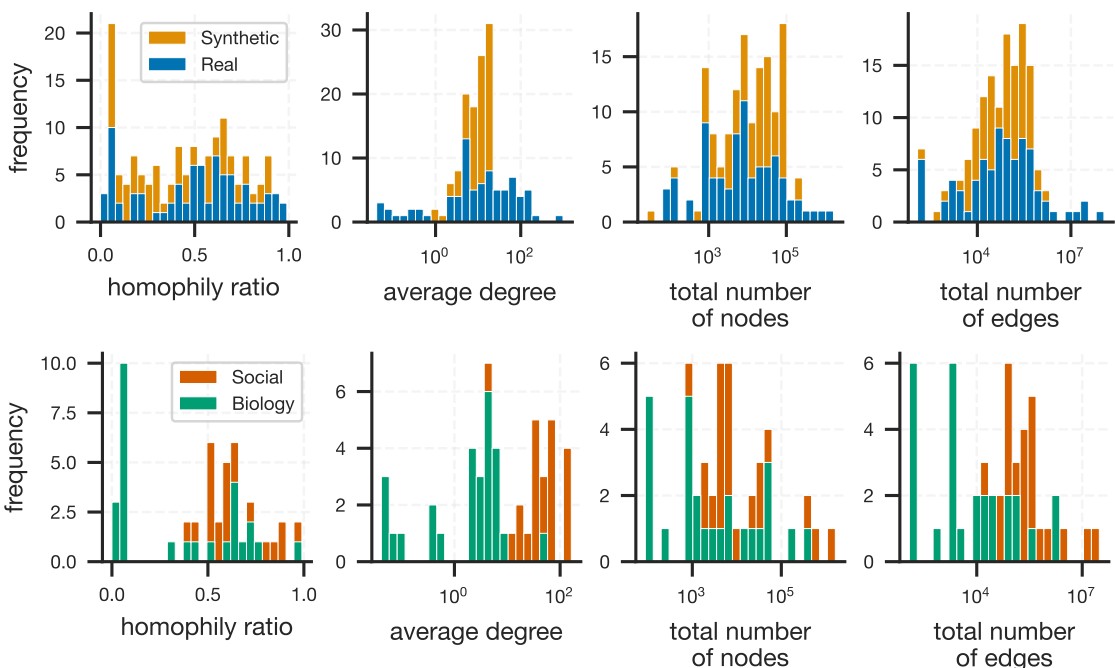

Figure A4: **Characteristics of graph datasets used to train GraphFM:** From left to right, we compute the histograms of the homophily ratio, average degree, number of nodes and number of edges of all 152 graphs used during training. The homophily ratio provides a measure of how frequently a node is directly connected to other nodes from the same class.

## D.2 Scaling analysis breakdown for different test datasets

The main text reports the average effect of scaling. In Figure A5, we provide a dataset-level breakdown. While all datasets benefit from increased model and data scale, the magnitude of improvement varies, with more challenging datasets (e.g., Chameleon) showing larger relative gains compared to easier ones (e.g., Coauthor-Physics).

## D.3 Ranking of different models

We compare the mean rank of GraphFM against specialist baselines across the 10 out-of-distribution datasets (Figure A6). Unlike baseline models that require extensive hyperparameter tuning for each dataset, we evaluated GraphFM using a fixed configuration (learning rate $= 10^{-3}$) for both MFT and NFT strategies. For NFT, we additionally applied a simple unfreezing schedule (Appendix A.3.2).

As shown in Figure A6, NFT achieves the best overall rank, while MFT achieves the second-best rank with the lowest variance. Specialist models such as H2GCN and NAG show higher variability in rank due to their specialization for heterophilic and homophilic graphs, respectively. A detailed per-dataset comparison is provided in Appendix B.6.

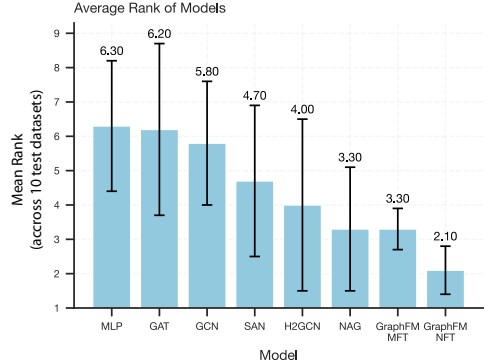

Figure A6: Mean rank of various models accross 10 unseen test datasets (lower is better).

## D.4 Generalization to Graph Classification

To further evaluate the generalization of GRAPHFM, we conducted additional experiments on graph-level classification using molecular datasets—MUTAG and PROTEINS. In this setting, we extend the GRAPHFM architecture by adding a cross-attention–based graph decoder following the Perceiver encoder.

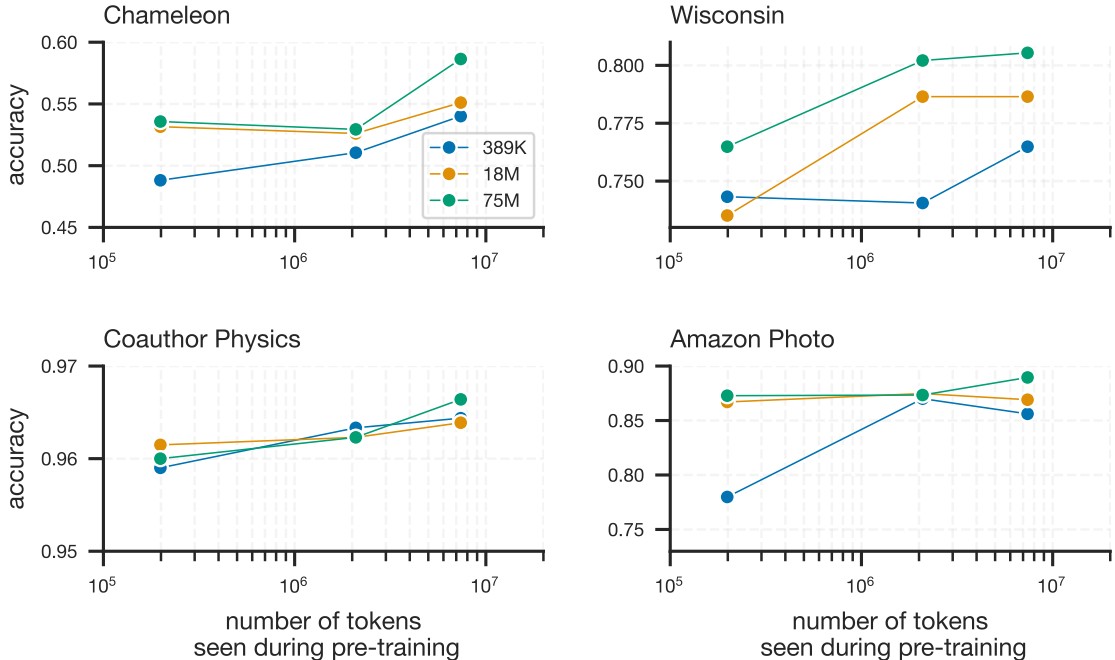

Figure A5: Accuracy as the model and dataset size are increased. Results are shown for four datasets, Chameleon and Wisconsin (heterophilic), and Coauthor Physics and Amazon Photo (homophilic).

This decoder aggregates information from the latent tokens to produce graph-level representations, which are then used for label prediction.

We apply MLP fine-tuning while keeping the cross-attention decoder learnable. Table A7 shows GRAPHFM achieves substantial improvements over existing graph transformer and message-passing baselines, demonstrating strong adaptability to graph-level prediction tasks.

We note that the current version of GRAPHFM does not handle edge features, which is why MUTAG and PROTEINS were chosen for this experiments. Nevertheless, the strong results on these benchmarks indicate that the model architecture is capable of generalizing to graph-level classification.

Table A7: Graph classification results for GRAPHFM (baselines reported from (Guan et al.)).

| Model | MUTAG | PROTEINS |
|---|---|---|
| GCN | $74.6 \pm 7.7$ | $73.1 \pm 3.8$ |
| GraphSAGE | $74.9 \pm 8.7$ | $73.8 \pm 3.6$ |
| GT | $75.5 \pm 7.9$ | $68.4 \pm 3.3$ |
| Graphormer | $74.4 \pm 7.4$ | $68.4 \pm 3.7$ |
| SAT | $81.4 \pm 8.2$ | $71.7 \pm 3.5$ |
| SpecFormer | $83.7 \pm 8.0$ | $72.0 \pm 4.9$ |
| **GraphFM (ours)** | $\mathbf{93.8 \pm 6.6}$ | $\mathbf{76.4 \pm 3.7}$ |

### D.5 Additional Baselines

The main text presents a comparison of GRAPHFM with baselines that are more consistently reported across the literature. Table A8 and Table A9 provides additional baselines for all the unseen test datasets.

### D.6 GraphAny Comparisons

GraphAny is primarily designed for zero-shot transfer, whereas our focus with GraphFM is on supervised and fine-tuned performance across diverse benchmarks. As shown in Table A10, GraphAny's absolute accuracy remains significantly below that of fine-tuned methods. On average, GraphFM outperforms GraphAny by 7.11% across datasets. For example, on *ogbn-arxiv*, the best GraphAny model achieves 58.68% accuracy, compared to 70.01% with GraphFM after fine-tuning. These results highlight that while GraphAny demonstrates potential for zero-shot settings, substantial performance gains can still be obtained through multi-graph pretraining and task-specific adaptation with GraphFM.

Table A8: Results on node classification tasks for large graph datasets. We report the accuracy (%) with standard deviation over 10 splits (OOM indicates Out of Memory).

| Method | Photo | Physics | CS | ogbn-arxiv | Comp |
|---|---|---|---|---|---|
| **GCN-based methods** | | | | | |
| GCN (Jiang et al., 2019) | 85.94±1.18 | 95.38±0.20 | 94.06±0.16 | 70.40±0.10 | 89.47 ± 0.46 |
| GGCN (Yan et al., 2022) | 57.84±14.6 | 95.89±0.21 | 89.94±2.24 | 62.71±1.76 | - |
| APPNP (Klicpera et al., 2019) | 84.71±1.25 | 95.04±0.31 | 87.49±0.48 | 70.20±0.16 | 90.18 ± 0.17 |
| GCNII (Chen et al., 2020) | 67.06±1.74 | 94.88±0.32 | 84.23±0.78 | 69.78±0.16 | - |
| GAT (Velickovic et al., 2017) | 87.13±1.00 | 95.14±0.28 | 93.61±0.14 | 67.56±0.12 | 90.78 ± 0.13 |
| GATv2 (Brody et al.) | 81.52±3.23 | 95.02±0.32 | 88.46±0.61 | 68.84±0.13 | - |
| SuperGAT (Kim & Oh) | 85.83±1.29 | 95.11±0.26 | 88.11±0.43 | 66.99±0.07 | - |
| **Heterophily-based methods** | | | | | |
| MLP (LeCun et al., 2015) | 88.66±0.85 | 95.12±0.26 | 92.99±0.51 | 52.63±0.12 | 84.63 |
| MixHop (Abu-El-Haija et al., 2019) | 93.24±0.59 | 96.34±0.22 | 93.88±0.63 | 70.83±0.30 | - |
| H2GCN (Jing et al., 2024) | 91.56±0.70 | 96.28±0.13 | 94.02±0.31 | 68.29±0.67 | 89.33 ± 0.27 |
| FAGCN (Bo et al., 2021) | 87.53±0.75 | 95.86±0.12 | 91.82±0.54 | 66.12±0.02 | - |
| GPRGNN (Chien et al., 2021) | 92.27±0.44 | 96.06±0.21 | 93.60±0.36 | 68.28±0.21 | 89.32 ± 0.29 |
| **Graph Transformer-based methods** | | | | | |
| SAN (Kreuzer et al., 2021) | 94.17±0.65 | 96.83±0.18 | 94.16±0.36 | 69.17±0.15 | 89.83 ± 0.16 |
| Graphormer (Ying et al., 2021) | 85.20±4.12 | OOM | OOM | OOM | OOM |
| LiteGT (Chen et al., 2021) | - | OOM | 92.16±0.44 | OOM | - |
| UniMP (Wang et al., 2025) | 92.49±0.47 | 96.82±0.13 | 94.20±0.34 | 73.19±0.18 | - |
| DET (Guo et al., 2022) | 91.44±0.49 | 96.30±0.18 | 93.34±0.31 | 55.70±0.30 | - |
| NAGphormer (Chen et al.) | 94.64±0.60 | 96.66±0.16 | 95.00±0.14 | 68.21 ± 0.021 | 91.22 ± 0.14 |
| GRAPHFM -MFT | 93.01±1.82 | 96.64±0.17 | 95.19±0.21 | 65.29±0.16 | 89.95 ± 0.83 |
| GRAPHFM -NFT | 94.37±0.35 | 96.77±0.12 | 95.24±0.18 | 70.01±0.18 | 90.07 ± 0.21 |

Table A9: Results on node classification tasks for heterophilic graphs. We report the test accuracy across many heterophilic graph benchmark datasets. The standard deviation is reported across 10 train/test splits.

| Method | Texas | Wisconsin | Actor | Squirrel | Chameleon |
|---|---|---|---|---|---|
| **GCN-based methods** | | | | | |
| GCN (Jiang et al., 2019) | 55.14 ± 5.16 | 51.76 ± 3.06 | 27.32 ± 1.10 | 31.52 ± 0.71 | 38.44 ± 1.92 |
| GAT (Velickovic et al., 2017) | 52.16 ± 6.63 | 49.41 ± 4.09 | 27.44 ± 0.89 | 36.77 ± 1.68 | 48.36 ± 1.58 |
| GraphSAGE (Hamilton et al., 2017a) | 82.43 ± 6.14 | 81.18 ± 5.56 | 34.23 ± 0.99 | 41.61 ± 0.74 | 58.73 ± 1.68 |
| **Heterophily-based methods** | | | | | |
| MLP (LeCun et al., 2015) | 80.81 ± 4.75 | 85.29 ± 3.31 | 36.63 ± 0.70 | 28.77 ± 1.56 | 46.21 ± 2.99 |
| HH-GCN (Azabou et al., 2023b) | 71.89 ± 3.46 | 79.80 ± 4.30 | 35.12 ± 1.06 | 47.19 ± 1.21 | 60.24 ± 1.93 |
| HH-GAT (Azabou et al., 2023b) | 80.54 ± 4.80 | 83.53 ± 3.84 | 36.70 ± 0.92 | 46.35 ± 1.86 | 61.12 ± 1.83 |
| HH-GraphSAGE (Azabou et al., 2023b) | 85.95 ± 6.42 | 85.88 ± 3.99 | 36.82 ± 0.77 | 45.25 ± 1.52 | 62.98 ± 3.35 |
| MixHop (Abu-El-Haija et al., 2019) | 77.84 ± 7.73 | 75.88 ± 4.90 | 32.22 ± 2.34 | 43.80 ± 1.48 | 60.50 ± 2.53 |
| GGCN (Yan et al., 2022) | 84.86 ± 4.55 | 86.86 ± 3.29 | 37.54 ± 1.56 | 55.17 ± 1.58 | 71.14 ± 1.84 |
| H2GCN (Jing et al., 2024) | 84.86 ± 7.23 | 87.65 ± 4.98 | 35.70 ± 1.00 | 36.48 ± 1.86 | 60.11 ± 2.15 |
| LINKX (Lim et al., 2021) | 74.60 ± 8.37 | 75.49 ± 5.72 | 36.10 ± 1.55 | 61.81 ± 1.80 | 68.42 ± 1.38 |
| **Graph Transformer-based methods** | | | | | |
| SAN (Kreuzer et al., 2021) | 60.17 ± 6.66 | 51.37 ± 3.08 | 27.12 ± 2.59 | 39.92 ± 2.14 | 44.32 ± 1.73 |
| UniMP (Wang et al., 2025) | 73.51 ± 8.44 | 79.60 ± 5.41 | 35.15 ± 0.84 | - | - |
| NAGphormer (Chen et al.) | 63.51 ± 5.85 | 62.55 ± 6.22 | 34.33 ± 0.94 | 49.93 ± 0.07 | 57.39 ± 0.02 |
| Gapformer (Liu et al., 2023) | 80.27 ± 4.01 | 83.53 ± 3.42 | 36.90 ± 0.82 | - | - |
| GRAPHFM -MFT | 80.81 ± 2.76 | 83.13 ± 2.35 | 36.29 ± 0.63 | 42.80 ± 1.54 | 58.64 ± 1.24 |
| GRAPHFM -NFT | 82.16 ± 3.24 | 83.62 ± 3.21 | 38.01 ± 1.07 | 42.98 ± 1.62 | 59.12 ± 1.64 |

Table A10: Comparison of GraphFM and GraphAny on node classification datasets.

| Dataset | GCN | MLP | GAT | GraphAny (Arxiv) | GraphAny (Wisconsin) | GraphFM (MFT) | GraphFM (NFT) |
|---|---|---|---|---|---|---|---|
| Comp | $58.82 \pm 2.98$ | $85.83 \pm 0.86$ | $87.01 \pm 0.50$ | $83.04 \pm 1.24$ | $82.09 \pm 1.22$ | $95.13 \pm 0.45$ | $95.44 \pm 0.47$ |
| Photo | $68.20 \pm 0.88$ | $91.88 \pm 0.79$ | $91.86 \pm 1.07$ | $90.60 \pm 0.82$ | $90.18 \pm 0.91$ | $93.01 \pm 1.82$ | $94.37 \pm 0.35$ |
| CS | $85.88 \pm 0.93$ | $81.83 \pm 0.71$ | $88.47 \pm 0.79$ | $90.45 \pm 0.59$ | $90.85 \pm 0.63$ | $95.10 \pm 0.21$ | $95.24 \pm 0.18$ |
| Physics | $87.43 \pm 1.98$ | $93.93 \pm 0.37$ | $93.01 \pm 0.89$ | $92.69 \pm 0.52$ | $92.54 \pm 0.43$ | $96.54 \pm 0.17$ | $96.77 \pm 0.12$ |
| Arxiv | $55.50 \pm 0.23$ | $71.74 \pm 0.29$ | $73.65 \pm 0.11$ | $58.68 \pm 0.17$ | $57.79 \pm 0.56$ | $69.96 \pm 0.21$ | $70.01 \pm 0.18$ |
| Chameleon | $36.62 \pm 0.87$ | $64.69 \pm 2.21$ | $67.76 \pm 0.72$ | $62.59 \pm 0.86$ | $60.09 \pm 1.93$ | $58.64 \pm 1.24$ | $59.12 \pm 1.61$ |
| Squirrel | $30.36 \pm 0.78$ | $47.07 \pm 0.71$ | $46.69 \pm 1.44$ | $46.70 \pm 0.95$ | $42.34 \pm 3.46$ | $42.80 \pm 1.54$ | $42.98 \pm 1.62$ |
| Texas | $48.65 \pm 4.01$ | $31.55 \pm 2.71$ | $50.45 \pm 2.41$ | $72.97 \pm 2.71$ | $73.51 \pm 1.21$ | $80.51 \pm 2.76$ | $82.16 \pm 3.24$ |
| Wisconsin | $66.67 \pm 5.31$ | $37.25 \pm 1.64$ | $52.94 \pm 3.18$ | $71.77 \pm 5.66$ | $71.18 \pm 5.08$ | $70.92 \pm 1.52$ | $73.63 \pm 1.87$ |
| Actor | $33.95 \pm 0.80$ | $28.55 \pm 0.68$ | $27.30 \pm 0.22$ | $28.60 \pm 0.21$ | $29.51 \pm 0.55$ | $36.29 \pm 0.63$ | $38.01 \pm 1.07$ |

