# OpenReview forum: "GraphFM: A generalist graph transformer that learns transferable representations across diverse domains"
_TMLR — Accepted by TMLR_

### Review · Reviewer_SgFC · 2025-09-25

**Summary Of Contributions:**

This paper introduces GraphFM, a graph-based transformer dedicated to learn from various graph domains (e.g., citation graph, recommendations, molecules, etc.) and to use the resulting model as a pretrained foundation model to achieve node-prediction tasks on graphs from unseen domains. Three contributions are claimed in the paper:

1. A dedicated algorithm to efficiently dispatch graphs on GPUs during the pre-training phase.
2. The per-se generalist foundation model able to transfer knowledge to unseen graph. Briefly, the main idea is to leverage latent query tokens allowing a communication between distant nodes, which is challenging in standards graph neural network architecture. The remaining of the architecture (transformer with attention layers, encoding and decoding blocks) is relatively standard.
3. An analysis of the performance of the models on different aspects (comparison with dedicated approaches, stability regarding the hyper-parameters, efficiency of the GPU-dispatch, etc.)

---
**List of strengths:**

1. The paper addresses a challenging problem to export the idea of foundation models to graph-related tasks. The topic is definitively appropriate for TMLR.
2. To my knowledge, the methodology is sound and the contribution is well grounded regarding the related work.
3. The experimental protocol is mostly sound and promising results are reported. I only have a few weaknesses to report concerning the experiments (detailed below).
4. The paper is well-written and very pleasant to read.

---
**List of weaknesses:**

1. The paper focuses on node-level prediction and only features node features. As genericity among graph-related tasks is the main goal of the contribution, it is a missed opportunity to do not consider features on the edges, or graph-level/edge-level predictions. On this aspect, future work is still required to reach this full genericity.
2. I am surprized that there is no source code provided, nor explanation that it is planned to do. Without the implementation available,  the impact of this paper is severely decreased.
3. It is not clear why other graph foundation modeling models (e.g., Mao et al., 2024, properly referenced in the paper) could not be used for the tasks proposed. It would have been interesting to have a comparison with this approach, or to explain why this comparison is not relevant.
4. The bibliography needs to be significantly refactored. There are duplicated entries (Fey and Lenssen, 2019), missing information (e.g., Chen et al.), ArXiv references instead of the published one (Dao, 2023), and incorrect capitalization (Gfse).

**Additional Comments:**

I am not asking to go beyond the node level as it will severely change the paper and as the current paper can still be of interest of the audience. However, I still think that not considering edge features will decrease the global impact of the paper.

It will be very interesting to see how the model perform on combinatorial problems over graphs (e.g., the Travelling salesman problem). Benchmarks are provided here (https://github.com/graphdeeplearning/benchmarking-gnns)

**Audience:**

Yes

**Audience Explanation:**

**This needs further clarification.**

In terms of findings, the main conclusion that it is possible to train foundation models on graph was already known (properly discussed in the paper) – so the main interest of this paper is on the realization/implementation side.
For that, this paper will be of interest for the research community related on learning on/from graphs. However, not having the source code available severely decreases the interest/impact that we can have on this paper. This needs to be clarified.

**Broader Impact Concerns:**

I have nothing specific to report here

**Claims And Evidence:**

Yes

**Claims Explanation:**

**The claims are mostly supported.** The methodology, although incremental, is sound. Experiments proposed confirms the hypothesis that we can train a foundation model on graphs. The main weakness is that the architecture only consider node-related features and node-level predictions decreases the genericity of the model, and consequently the main claim. There are also some (minor) missing experiments/information to be fully convinced about the benefit of this architecture.

1. Figure 3: Why not comparing SSSampler with 8 GPUs as well (as for the baseline)?
2. A comparison with Mao et al. 2024 is missing. Is this comparison is not relevant, please explain why in the paper.
3. Experiments Q2: It will be interesting to discuss the number of parameters of the specialized approaches. I guess that there are much lower than the ones of the foundation model.

**Requested Changes:**

**Major changes:**
1. Refactoring the bibliography (see my previous comments).
2. Providing the source code or explaining why it is not possible.
3. Adding a comparison with 8 GPUs on Figure 3 for DistributedSSSampler.
4. Adding the number of parameters of specialized approaches in experiment Q2.
5. Adding a comparison with Mao et al. 2024 – or explaining why it is not relevant.

**Typos:**
1. Page 2: “GraphFM ,” -> “GraphFM,”
2. Page 4: it’s -> its

---

> ### Author Response · Authors · 2025-10-10
> **Response to Reviewer**
>
> Thank you for your time and thoughtful review. We are glad that you found the paper “well-written and very pleasant to read” and that “the methodology is sound and the contribution is well grounded regarding the related work.” We have incorporated most of the major requested changes in the revised PDF (highlighted in blue) and provide point-by-point responses to your comments below.
>
> **Major Changes:**
>
> **\[Q1\]:** *“Refactoring the bibliography”*
>
> **Reply:** Thank you for pointing this out. We have refactored the bibliography by removing duplicate entries and replacing arXiv citations with published versions where available.
>
> **\[Q2\]:** *“Providing the source code or explaining why it is not possible.”*
>
> **Reply:** We apologize for not including the source code. We have now updated the supplementary material to include the source code for pretraining the model.
>
> **\[Q3\]:** *“Adding a comparison with 8 GPUs on Figure 3 for DistributedSSSampler.”*
>
> **Reply:** For the N \= 64 point in Figure 3, we do in fact use 8 GPUs. Here, N refers to the number of buckets (interpreted as the number of *virtual* GPUs), computed as num\_gpus × num\_grad\_accumulation\_steps. In this case, we used 8 GPUs with 8 gradient accumulation steps. Since our method depends on the number of buckets rather than the number of GPUs, the same effect can also be achieved on a single GPU with 64 gradient accumulation steps, or with 64 GPUs and 1 gradient accumulation step. We apologize for the confusion in our earlier explanation, we have updated the Figure 3 caption (page 6\) to clarify this.
>
> **\[Q4\]:** *“Adding the number of parameters of specialized approaches in experiment Q2.”*
>
> **Reply:** Thank you for the suggestion. It is challenging to report parameter counts for all models and datasets, as many prior works do not provide complete configurations or disclose exact hyperparameters. However, for reference, we estimated the number of parameters for NAG, the best-performing specialist model in our comparison, based on its released implementation and reported hyperparameters.
>
> On the Computers dataset, NAG has approximately 2.6M parameters, while on CS it has around 6.8M. Taking a conservative average of 4M parameters per dataset, and considering that we evaluate on 10 datasets, this amounts to roughly 40M total parameters. In comparison, our pretrained model has approximately 75M parameters. While specialized models are smaller in isolation, their overall cost scales linearly with the number of datasets and can increase substantially when accounting for hyperparameter tuning. In contrast, the amortized cost of a single pretrained model is much lower in the long run, as it can be reused and adapted for new datasets.
>
> **\[Q5\]:** *“Adding a comparison with Mao et al. 2024 – or explaining why it is not relevant.”*
>
> **Reply:** To ensure we are addressing the correct work, could you kindly confirm the exact reference? The paper by Mao et al. (2024) that we were able to locate is *“Position: Graph Foundation Models Are Already Here”*, which is a position piece outlining guiding principles for GFMs and categorizing current efforts. We have already cited this work in our manuscript. Many of the models discussed there, such as ULTRA (targeted to knowledge graphs) or OFA (which relies on converting graph features into text for LLMs), are designed for specific domains or homogenize the feature space through LLMs, making them not directly comparable to our setting.
>
> **Typos:**
>
> **\[Q1\]:** *“Page 2: “GraphFM ,” \-\> “GraphFM,” and Page 4: it’s \-\> its”*
>
> **Reply:** Thank you for pointing this out. We have corrected these typos in the revised PDF.
>
> **Additional Comments:**
>
> **\[Q1\]:** *“I am not asking to go beyond the node level as it will severely change the paper and as the current paper can still be of interest of the audience. However, I still think that not considering edge features will decrease the global impact of the paper.”*
>
> **Reply:** We agree that incorporating edge features could substantially broaden the impact of our work. However, our current study is intentionally scoped to heterogeneity in labels and node features in order to maintain a focused contribution and controlled evaluation. Extending the framework to handle edge features would require non-trivial architectural modifications, which we consider an important direction for future work.
>
> **\[Q2\]:** *“It will be very interesting to see how the model perform on combinatorial problems over graphs (e.g., the Travelling salesman problem). Benchmarks are provided here (https://github.com/graphdeeplearning/benchmarking-gnns)”*
>
> **Reply:** We agree that evaluating on combinatorial problems would be interesting. However, these are formulated as edge-level or link prediction tasks, which fall outside the scope of the current paper where we focus on node-level prediction. Extending our framework to edge-level tasks is an exciting direction for future work.

---

> ### Comment · Reviewer_SgFC · 2025-10-20
>
> Dear authors,
>
> Many thanks for the detailed answer. It resolves most of my concerns. Please also make clear your answer to Q4 in your revised version.
>
> For the comparison of Q5 - you are right. The wording used in the paper (e.g., *recent efforts in graph neural networks (GNNs) have shown success in training models on many graphs (Beaini et al., 2024; Mao et al., 2024)*, suggested it was a competitor. Please refactor this in order to avoid misunderstanding.
>
> There are still formatting issues in the bibliography. For instance,
>
> > Arnab Sinha, Zhihong Shen, Yang Song, Hao Ma, Darrin Eide, Bo-June Hsu, and Kuansan Wang. An
> > overview of microsoft academic service (mas) and applications. In Proceedings of the 24th international
> > conference on world wide web, pp. 243–246, 2015
>
> is not appropriately capitalized (mas -> MAS).
>
> Otherwise, I am satisfied with the rebuttal. My main remaining concern (which seems shared by the other reviewers) is the limitation to the node level - but again, I do not think there is something that can be done in the current paper (and I am not requesting that).

---

### Review · Reviewer_gPND · 2025-09-27

**Summary Of Contributions:**

This paper contributes a scalable graph pre-training approach based on a perceiver-based transformer encoder. The paper demonstrates its benefits on vast pretraining on over 150 diverse datasets including homophilous and heterophilous graphs. While previous works only aspire to create graph foundation models on specific sub domains (e.g. MoleBert), there are no large scale generalist pretrained graph foundation models to date.

## Strengths

1. I appreciate the discussion of heterophilous and homophilous graph types and its usage as a guide for the construction of the dataset, the dataset level statistics, and the thoroughness detailed in appendix B
2. Figure 1 is well made and served as a helpful aid during my comprehension of this paper
3. Authors pay delicate attention to training optimizations, innovating using a novel snake allocation procedure and using FlashAttention to speed up their training
4. The experimental section has a clean recursive structure alternative between well defined questions and experiments that attempt to answer them. The results from Figure 5 provide compelling evidence that their methodology in this paper has great promise

## Weakness

1. There are places where the exposition can be improved, which I have highlighted throughout my requested changes. In particular, this paper would benefit from a background section.
2. Some figures and tables can be changed to better visualize the experimental results
3. The categories "Soc", "Bio", and "All" seem a bit arbitrary to study generalization across domains

**Additional Comments:**

1. Would this approach work for graphs that contain both node features and positions? Do any of the datasets include molecule graphs that include 3D coordinates? Is this an avenue for future work?
2. While not strictly necessary, the authors may consider adoptng the datasets proposed in [1], Lim et al. above
3. Similarly, while not strictly necessary, I think the authors should include LINKX from [1] as a baseline and potentially also Unitary Convolutions [4], which is also suited for heterophilous graphs. The latter may be expensive to compute all of the exponentiated adjacency matrices, which probably isn't worth the effort considering there are similar baselines already
4. Why do the authors not consider regression? Is it just to keep the study well scoped and the ability to compute the homophily ratio? It would be nice if a couple of regression tasks were included, even if to much a less extent.

[4] https://arxiv.org/abs/2410.05499

**Audience:**

Yes

**Audience Explanation:**

This paper is interesting to the graph learning community both for the general pre-training methods and the favorable results on down stream tasks. The comparison of different methods of both heterophilous and homophilous graphs is of further independent interest and elucidates how different inductive bias in different architectures are more favorable on one class of problems than another.

**Broader Impact Concerns:**

Following section 5.7 in [3], it would be great if the authors could estimate the Carbon footprint of large scale graph pretraining and discuss in a broader impacts statement

[3] https://openreview.net/pdf?id=KoFOg41haE

**Claims And Evidence:**

Yes

**Claims Explanation:**

The core contributions of this paper, in the authors words, are
1. Scalable Pre-training
2. Demonstration of Benefits from Across-Graph Pre-training
3. Scaling Analysis and Impact of Multi-Graph Pre-training

The authors also pose and answer various subquestions in the experimental section that aid in addressing these contributions. My main concern would be with the claim of the second contribution. I think the categories of "Soc", "Bio", and "All" are slightly arbitrary. In the last paragraph before Q2, I would add this caveat to hedge the result more. Beyond that, the ablations on GPU utilization and training time as well as fine-tuning strategy, the accuracy of their graphs for classification, and thoroughness of dataset curation all support claims made in the paper

**Requested Changes:**

## Major / Intermediate

1. This paper would benefit from a short background section that can help the reader understand the design choices and methodology throughout section 2. Please provide a background section on the necessary topics necessary to understand the paper. This would also help clarify what is being contributed in the current paper
2. The violin plots in figure 3 are hard to compare and can be replaced with overlapping histograms with some opacity threshold and color labeling
3. In datasets, the authors should compute the straightforward alternative homophily metric in eqns 2 and 3 in [1], which essentially captures the excess homophily above a randomly wired graph
4. "the normalized l2-distance of their hyperparameter vectors to the optimal hyperparameter vector." Is this just lr and weight decay? The full set of items considered hyperparameters in the vector should be spelled out. It is not immediately clear to me. This may not make sense if the hyperparams tend to be of different order of magnitude as well.
5. Some more justification for the "Soc", "Bio", and "All" categories is in order, and potentially the authors should qualify their results in this context

[1] https://arxiv.org/pdf/2110.14446

## Minor

1. In section 2.1.1, the authors refer to the positional embedding of the node. On first read, it wasn't clear to me that the authors meant a positional encoding that correspondeded to the order in which the node feature was fed into the model and not cartesian coordinates in 3D space, which might be considered for many atomic datasets. Please make this distinction clearer, or begin the paper with a short background section detailing various aspects of the tokenization process in the literature.
2. The authors use SignNet [2] to help construct the positional encodings. This is well motivated by Proposition 4 and Figure 2 in [2]. However, the authors do not comment on why they use SignNet here. The authors should add a few words justifying their design choice here, since it may not be readily known to the audience
3. ***Extremely nitpicky***, but I don't like the use of the word "Remark" for the paragraph under Figure 2. That kind of organization helps in the mathematics literature for well defined mathematical statements that don't constitute full theorems. Please start this paragraph differently. The "remark" block appears later as well
4. Table A6 should include also the Avg Rank for Homophilic and Heterophilic graphs specifically, I would value this as a summary statistic
5. Tables A7 and A8 would benefit from a citation column allowing the interested reader to learn more about the purported baseline methods more easily. Please add this

[2] https://arxiv.org/pdf/2202.13013

---

> ### Author Response · Authors · 2025-10-10
> **Response to Reviewer's Major Comments**
>
> Thank you for your time and thoughtful review. We are glad that you appreciated our discussion of homophilous and heterophilous graphs, as well as our attention to training optimizations and experimental structure. We have incorporated the major requested changes in the revised PDF (highlighted in blue) and provide point-by-point responses below.
>
> **Major Comments:**
>
> **\[Q1\]**: *“This paper would benefit from a short background section that can help the reader understand the design choices and methodology throughout section 2\. Please provide a background section on the necessary topics necessary to understand the paper. This would also help clarify what is being contributed in the current paper”*
>
> **Reply:** Thank you for the helpful suggestion. We have added a more detailed background section to provide the necessary context for understanding our design choices. Please see Section 2 (page 2 and 3\) of the revised PDF for these additions.
>
> **\[Q2\]:** *“The violin plots in figure 3 are hard to compare and can be replaced with overlapping histograms with some opacity threshold and color labeling”*
>
> **Reply:** Thank you for the suggestion. We have updated Figure 3 (page 6).
>
> **\[Q3\]:** “*In datasets, the authors should compute the straightforward alternative homophily metric in eqns 2 and 3 in \[1\], which essentially captures the excess homophily above a randomly wired graph”*
>
> **Reply:** Thank you for the suggestion. We have incorporated the excess homophily metric, as an additional column in the dataset summary (Table A3) of the appendix (page 21).
>
> **\[Q4\]:** *"the normalized l2-distance of their hyperparameter vectors to the optimal hyperparameter vector." Is this just lr and weight decay? The full set of items considered hyperparameters in the vector should be spelled out. It is not immediately clear to me. This may not make sense if the hyperparams tend to be of different order of magnitude as well.”*
>
> **Reply:** The set of hyperparameters considered for GCN and our model differ, with GCN requiring many more  such as the number of hidden units, layers, and dropout rate to achieve good performance (see Table A5 for the complete search space). This is precisely why we chose the normalized L2 distance as our metric, as it allows comparison relative to each model’s own optimal configuration without being biased by the number of hyperparameters. We agree that the hyperparameters can vary in magnitude, which is why we normalized them within their respective ranges before computing the distance.
>
> **\[Q5\]:** *“Some more justification for the "Soc", "Bio", and "All" categories is in order, and potentially the authors should qualify their results in this context”*
>
> **Reply:** Thank you for this thoughtful comment. When deciding how to split the full dataset into smaller subsets for investigation, we found that there were two main types of graphs, “Soc” which are social networks and recommendation network graphs, “Bio” which are molecular graphs, and then “All” encompassed the rest of the graphs that didn’t have a easy category. In all three sets, we ensured that there were a sufficiently large number of datasets, which enabled us to perform a meaningful analysis without bias in data sizes across the three. We acknowledge that a more comprehensive domain-wise study would be required to make stronger claims. In this work, our goal was to provide preliminary evidence that including datasets from seemingly unrelated domains can improve performance. We have added this clarification and caveat in the last paragraph before Q2 (page 8). Please refer to the updated PDF.
>
> ---
>
> [1] Lim, D., Hohne, F., Li, X., Huang, S. L., Gupta, V., Bhalerao, O., & Lim, S. N. (2021). Large scale learning on non-homophilous graphs: New benchmarks and strong simple methods. Advances in neural information processing systems, 34, 20887-20902.

---

> ### Author Response · Authors · 2025-10-10
> **Response to Reviewer's Minor and Additional Comments**
>
> **Minor Comments:**
>
> **\[Q1 and Q2 \]:** “*In section 2.1.1, the authors refer to the positional embedding of the node. On first read, it wasn't clear to me that the authors meant a positional encoding that corresponded to the order in which the node feature was fed into the model and not cartesian coordinates in 3D space, which might be considered for many atomic datasets. Please make this distinction clearer, or begin the paper with a short background section detailing various aspects of the tokenization process in the literature.”*
>
> The authors use SignNet \[2\] to help construct the positional encodings. This is well motivated by Proposition 4 and Figure 2 in \[2\]. However, the authors do not comment on why they use SignNet here. The authors should add a few words justifying their design choice here, since it may not be readily known to the audience
>
> **Reply:** Thank you for the suggestion. We have added a more detailed background section to provide the necessary context for understanding our design choices. Please see Section 2 (page 2 and 3) of the revised PDF for these additions.
>
> **\[Q3\]:** *“Extremely nitpicky, but I don't like the use of the word "Remark" for the paragraph under Figure 2. That kind of organization helps in the mathematics literature for well defined mathematical statements that don't constitute full theorems. Please start this paragraph differently. The "remark" block appears later as well”*
>
> **Reply:** We have replaced all instances of “Remark” with “Note” in the revised version of the paper.
>
> **\[Q4\]:** *“Table A6 should include also the Avg Rank for Homophilic and Heterophilic graphs specifically, I would value this as a summary statistic”*
>
> **Reply:** Thank you for the suggestion. We have updated Table A6 (page 23) to include the average rank for homophilic and heterophilic graphs separately.
>
> **\[Q5\]:** *“Tables A7 and A8 would benefit from a citation column allowing the interested reader to learn more about the purported baseline methods more easily. Please add this”*
>
> **Reply:** Thank you for the suggestion. We have added a citation column to Tables A7 and A8 (page 28) .
>
> **Additional Comments:**
>
> **\[Q1\]:** *“Would this approach work for graphs that contain both node features and positions? Do any of the datasets include molecule graphs that include 3D coordinates? Is this an avenue for future work?”*
>
> **Reply:** Yes, our approach supports graphs with both node features and positional information. In this work, we use Laplacian-based embeddings via SignNet, which are compatible with any node feature space. The framework can also be extended to graphs with Euclidean positions, such as mesh-based datasets with 3D coordinates, by replacing SignNet with an MLP-based positional encoder. Since these datasets are primarily designed for graph-level classification, while our focus here is on node-level transfer across heterogeneous datasets, we leave this extension as an exciting direction for future work.
>
> **\[Q2\]**: *“While not strictly necessary, the authors may consider adopting the datasets proposed in \[1\], Lim et al. above”*
>
> **Reply:** We appreciate the suggestion. The datasets proposed by Lim et al. \[1\] are already included in our pretraining corpus (see Table A3), and therefore we cannot use them for evaluation.
>
> **\[Q3\]:** *“Similarly, while not strictly necessary, I think the authors should include LINKX from \[1\] as a baseline and potentially also Unitary Convolutions \[4\], which is also suited for heterophilous graphs. The latter may be expensive to compute all of the exponentiated adjacency matrices, which probably isn't worth the effort considering there are similar baselines already”*
>
> **Reply:** Thank you for the suggestion. We have included LINKX as a baseline in Table A8 (page 28). However, since results for homophilic datasets are not available for this method, we cannot include it in the overall rank computation.
>
> **\[Q4\]:** *“Why do the authors not consider regression? Is it just to keep the study well scoped and the ability to compute the homophily ratio? It would be nice if a couple of regression tasks were included, even if to much a less extent.”*
>
> **Reply:** Thank you for the suggestion. We limited our scope to node classification to maintain consistency across datasets and avoid task imbalance. Extending the framework to regression would require retraining the model and is an exciting direction for future work.
>
> ---
>
> [1] Lim, D., Hohne, F., Li, X., Huang, S. L., Gupta, V., Bhalerao, O., & Lim, S. N. (2021). Large scale learning on non-homophilous graphs: New benchmarks and strong simple methods. Advances in neural information processing systems, 34, 20887-20902.

---

> > ### Author Response · Authors · 2025-10-10
> > **Response to Reviewer's Broader Impact Concerns**
> >
> > **Broader Impact Concerns:**
> >
> > **\[Q1\]:** *“Following section 5.7 in \[3\], it would be great if the authors could estimate the Carbon footprint of large scale graph pretraining and discuss in a broader impacts statement”*
> >
> > **Reply:** Training our largest model took 1228.8 A40 hrs, totaling an energy usage of 368kWh. Assuming the GPUs were housed and powered at AWS's us-west-2 datacenter, with an carbon energy intensity of 0.25 kgCO2e/kWh and Power Usage Effectiveness of 1.15, the resulting emissions were approximately 106kg of CO2eq.
> > (To avoid accidentally disclosing our identity, we have used AWS as an example here. After the review process, we will update these with the kgCO2eq/kWh number at our actual location.)

---

> > > ### Comment · Reviewer_gPND · 2025-10-14
> > > **Response to authors changes**
> > >
> > > I appreciate the thought and efforts of the authors in changing the manuscript and addressing my concerns. In reading these responses as well as the comments left by other reviewers, I have a couple more global thoughts.
> > >
> > > One of my additional comments questioned the role of positional information in molecular graphs. What I was really getting at is the role of edge information (i.e. relative distances), in line with some of the concerns left by the other reviewers. I've gone back and forth on this in my head quite a bit. The paper already contributes a lot of useful insights toward scalable graph pre-training, and a method of introducing edge information could very well be its own follow up work. Still, I agree with the other reviewers that this is worth mentioning as a limitation of the current work. I also wonder if approaches like GINE [1] or GatedGCN [2] could be used to incorporate edge information in a simple manner, though this hypothesis is admittedly speculative and inexact.
> > >
> > > [1] https://arxiv.org/abs/1905.12265
> > > [2] https://arxiv.org/pdf/1711.07553
> > >
> > > In all, I am happy with the rebuttal and the changes that the authors have made. I am happy with the background as I think it clarifies the actual contributions better as well

---

> > > > ### Author Response · Authors · 2025-10-19
> > > > **Response to Reviewer**
> > > >
> > > > We are glad that you are happy with our rebuttal and the revisions to the manuscript. We agree with your observation regarding the incorporation of edge information, which is a key limitation of the current work. We will make sure to emphasize this limitation more clearly in the final version of the paper.

---

### Review · Reviewer_dSYo · 2025-10-06

**Summary Of Contributions:**

This work introduces a novel pretraining approach that learns from a variety of graph learning datasets to improve generalization capabilities on unseen datasets. GraphFM combines existing architectures from previous works to allow for multi task pretraining. The authors claim that GraphFM as a scalable framework with pretraining, reduces or eliminates the need for training on specific downstream tasks.

Furthermore, the authors introduce a new sampling method to improve GPU utilization and and sampling efficiency during pretraining.
Both contributions are supported by experimental results on various of downstream node classification datasets with GraphFM being scaled from 389K up to 75M parameters.

**Additional Comments:**

Strengths and Weaknesses:

Strengths:

S1 The paper is mostly clearly written and explained, the supporting figures and plots highlight the most important points.

S2 A good combination of sampling, architecture and pretraining into a single framework enabling concise pretraining across many datasets.

S3 Interesting pretraining mixture of datasets based on graph properties and synthetic graphs, one question regarding this: Did you consider other graph properties besides homophily while designing the pretraining precedure?

Weaknesses:

W1 Somewhat unclear reasoning for selection of PE (SignNet) and Encoder (Perceiver), with theoretical observations missing from the paper entirely.

W2 Limited gains from experiments, showcasing that the method is not stronger in empirical evaluation, despite substantial pretraining efforts made.

W3 Only limited selection for large-scale pretraining datasets (eg. no OGB, LRGB or other larger scale datasets) which limits the pretraining to smaller scale datasets and datasets of low quality [1].

W4 Very limited focus only on node classification and lacking experimental results for the claims of a generalist model, limited applicability because of missing tasks from edge- and graph-level tasks.

W5 Finetuning is necessary for each dataset, limiting the usefulness of the pretrained model compared to existing approaches as pretraining takes a significant amount of time and although fine tuning may be faster than training another model.

W6 Since the focus does not lie on novel methods here, the hyperparameter tuning and overall discussion of architecture choices remains very limited, thereby not really supporting the claims made by the authors.

[1] No Metric to Rule Them All: Toward Principled Evaluations of Graph-Learning Datasets, Coupette et al., ICML 2025

**Audience:**

Yes

**Audience Explanation:**

As pretraining on graph transformers has only been explored in few works before and few large scale pretraining objectives have been used in other architectures for graph learning, this work introduces a novel approach to pretraining which could be of interest. Additionally, the introduced sampling method could spark the audience's interest, as the experimental results indicate improved memory usage and utilization.

Furthermore, as results from other domains indicate, the importance of pretraining for large scale models is significant and is required for modern architectures.
Therefore, I believe that there could be suitable interest of TMLR's audience, especially in the graph learning domain.

**Broader Impact Concerns:**

There are no broader impact concerns from this work.

**Claims And Evidence:**

No

**Claims Explanation:**

One of the major contributions claimed by the authors is an improvement in generalization and results on downstream tasks. However, this claim is only partly supported by the experimental results shown in Sections 4, Appendix B.6, and D.5. As Table A6 shows, the proposed GraphFM model does not achieve state-of-the-art (SOTA) results on most datasets. Furthermore, GraphFM's results fall short compared to more recent methods, as seen in Tables A7 and A8.

However, the claimed improvements in sampling and GPU utilization are supported by the experimental results.
Because of this, it remains unclear whether the proposed improvements are supported by accurate evidence. Overall, the paper seems to be a limited contribution with unclear evidence for the claims made.

**Requested Changes:**

To strengthen the paper I would like the authors to include the following changes:

1. Provide clear intuition and explanation for the advantage on downstream tasks as claimed by the authors. In addition a clear motivation for the selected architecture should be given.

2. Fix unclear notation (eg. $\tilde{u}_i = MLP(u_i)$) and description of the concatenation of node features as described in section 2

3. Fix spelling mistakes such as “tthan” on the second to last line on page 3

4. Provide more experimental results to differentiate GraphFM from already existing models which do not require extensive pretraining and other graph transformer based methods.

---

> ### Author Response · Authors · 2025-10-17
> **Response to Reviewer dSYo (Regarding SOTA and Requested Changes)**
>
> We thank the reviewer for their detailed and constructive feedback.
>
> We would like to address one of the main concerns first \- that our model does not achieve SOTA results on most datasets. However, **our paper does not claim** that our model achieves SOTA performance. We, specifically, claim that our model is *competitive with SOTA*. In other words, we claim to achieve performances *close to* SOTA level, while maintaining our advantages of consistency, faster convergence, and hyperparameter stability. However, we understand that the phrase "competitive with SOTA" is subjective. We only mention this once in our paper (in the abstract) and once in the appendix. If you would prefer, we are happy to remove both mentions of SOTA from manuscript. We would like to maintain focus on the 4 major results that we explore in Section 5.2 of our paper---scaling analysis, consistency, faster convergence, and hyperparameter stability.
>
> Below, we provide point-by-point responses to your other questions and concerns. We have also incorporated the requested changes in the revised PDF (highlighted in blue).
>
> **Requested Changes:**
>
> **\[Q1a\]:** *“Provide clear intuition and explanation for the advantage on downstream tasks as claimed by the authors.”*
>
> **Reply:** The key advantage of GraphFM on downstream tasks is its faster convergence, stability, and low hyperparameter sensitivity. We provide detailed explanations and analyses of these advantages in Section 5.2 Q2, Q3, Q4. However, we are happy to present a summary of these advantages below:
>
> By pretraining across a diverse collection of graph datasets, GraphFM learns a shared latent representation that generalizes across domains, providing a strong initialization for fine-tuning. This allows the model to adapt efficiently to new datasets with minimal tuning while maintaining consistent performance across heterogeneous graph types.
> We refer the reviewer to Section 5.2 and our answer to Q4 in this rebuttal for more details, and would be happy to answer any further specific questions.
>
> **\[Q1b\]** *“In addition a clear motivation for the selected architecture should be given.”*
>
> **Reply:** Our choice of the Perceiver architecture was driven primarily by computational efficiency and scalability. Processing entire graphs with standard self-attention is infeasible due to quadratic cost, motivating the use of the Perceiver encoder which has significantly better time complexity (second last paragraph on page 4 of updated manuscript).
> Additionally, it is directly based on the standard attention operation, which allows us to leverage recent optimized attention kernel implementations (FlashAttention), making Perceiver an even more desirable option.
>
> Further details on other design components, including tokenization and the use of SignNet, are provided in Section 2 (Background) on page 2 and 3 in the revised version of the paper.
>
> **\[Q2\]:** *“Fix unclear notation (eg. $\tilde{u}\_i \= MLP(u\_i)$) and description of the concatenation of node features as described in section 2”*
>
> **Reply:** Thank you for the suggestion\! Reviewer gPND requested a more detailed background section, which addresses the issues you mentioned by clarifying the notation (e.g., $\\tilde{u}\_i \= \\mathrm{MLP}(u\_i)$) and the concatenation of node features, along with other related details. Please refer to Section 2 on pages 2–3 in the revised version of the paper.
>
> **\[Q3\]:** *“Fix spelling mistakes such as “tthan” on the second to last line on page 3”*
>
> **Reply:** Thank you for bringing this to our attention\! We have fixed the spelling error in the revised version (see page 4).
>
> Continued in the following response...

---

> > ### Author Response · Authors · 2025-10-17
> > **Continued Response to Reviewer dSYo (Requested Changes Continued and Additional Comments)**
> >
> > **Requested Changes (Continued)**:
> >
> > **\[Q4\]:** *“Provide more experimental results to differentiate GraphFM from already existing models which do not require extensive pretraining and other graph transformer based methods.”*
> >
> > **Reply:** We are unclear what you mean by "more experimental results." Section 5.2 (Q2-Q4) in our manuscript is heavily focused on differentiating our method from single-dataset graph models. We explain in detail in Section 5.2 (Q2-Q4), and are happy to summarize here, that our pretrained model has three desirable properties that single-dataset models don't have:
> >
> > * **Consistent performance across diverse graphs**
> >   * GraphFM achieves the best average rank with low variance across both homophilic and heterophilic graphs, as shown in Figure 5, demonstrating that it performs consistently well rather than excelling only on specific datasets.
> >   * Reviewer gPND also noted, *“the comparison of different methods on both heterophilous and homophilous graphs is of further independent interest and elucidates how different inductive biases in different architectures are more favorable on one class of problems than another.”*
> > * **Faster convergence and stability**
> >   * Compared to GCN and NAGphormer, GraphFM fine-tunes significantly faster and with greater stability, as shown in Figure 6\.
> >   * Reviewer WaZQ also noted, *“fine-tuning experiments under both MFT and NFT settings confirm robustness and stable convergence across tasks.”*
> > * **Lower hyperparameter sensitivity**
> >   * GraphFM maintains stable performance across a wide range of hyperparameter settings, as shown in Figure 7, demonstrating robustness to different hyperparameter choices
> >   * Reviewer WaZQ also noted, *“Figures 6 and 7 demonstrate markedly lower sensitivity to hyperparameters compared with GCN and NAGphormer.”*
> >
> > We believe our experiments have answered all the claims that we make. In the points above, we also included quotes from other reviewers who also found these claims justified. If you would like us to perform any particular experiment to answer a specific question, we would be happy to do so.
> >
> > ---
> >
> > **Additional Comments:**
> >
> > **\[S3\]:** *“Interesting pretraining mixture of datasets based on graph properties and synthetic graphs, one question regarding this: Did you consider other graph properties besides homophily while designing the pretraining precedure?”*
> >
> > **Reply:** Yes, for synthetic graphs, we also looked at the number of nodes in the graphs along with homophily. Specifically, we generated graphs that were uniformly distributed over these two axes \- homophily and number of nodes. For real graphs however, we just used all the graphs we could find.
> >
> > We would also like to share an additional result quantifying the contribution of synthetic datasets, conducted in response to reviewer WaZQ’s suggestion. As shown in Table R2, including synthetic datasets improves downstream performance, confirming that they provide useful structural diversity and help the model generalize across graph types.
> >
> > *Table R2: Mean downstream performance of the largest GraphFM model when pre-trained on all datasets versus excluding synthetic graphs.*
> >
> >
> > |  | Mean Performance |
> > | :---- | :---- |
> > | All Data | 0.7502 |
> > | All data excluding synthetic | 0.7345 |
> >
> > **\[W1\]:** *“Somewhat unclear reasoning for selection of PE (SignNet) and Encoder (Perceiver), with theoretical observations missing from the paper entirely.”*
> >
> > **Reply:** Reasoning for SignNet: We have added a background section (Section 2 on page 2 and 3\) clarifying our design choices. We use SignNet as our positional encoding because it offers a theoretically grounded approach to incorporating Laplacian eigenvectors while resolving their inherent sign ambiguity and preserving permutation invariance, which is essential for training across multiple graphs.
> >
> > Reasoning for Perceiver: Please refer to our response to Q1b above.
> >
> > **\[W2\]:** *“Limited gains from experiments, showcasing that the method is not stronger in empirical evaluation, despite substantial pretraining efforts made.”*
> >
> > **Reply:** We acknowledge that the absolute performance gains over existing models are modest, and our paper does not claim that GraphFM achieves the highest accuracy on every dataset. In our paper, we focus on the additional properties that emerge out of large-scaling pretraining. Please refer to our comment about achieving SOTA at the start of this rebuttal.
> >
> > Continued in next response...

---

> > > ### Author Response · Authors · 2025-10-17
> > > **Continued Response to Reviewer dSYo (Remaining Additional Comments)**
> > >
> > > **Additional Comments (Continued)**:
> > >
> > > **\[W3\]:** “*Only limited selection for large-scale pretraining datasets (eg. no OGB, LRGB or other larger scale datasets) which limits the pretraining to smaller scale datasets and datasets of low quality \[1\].”*
> > >
> > > **Reply:** Our pretraining mixture includes every dataset available from PyG and the Network Repository. We carefully avoided datasets that could overlap with evaluation datasets to avoid data leakage. For example, *Amazon2M* was intentionally removed because *Amazon Photos* and *Amazon Computers* are subsets of it. Similarly, *ogbn-papers100M* was excluded to prevent overlap with *ogbn-arxiv* (both of these are arxiv citation graphs), which is part of our evaluation set. We don't include LRGB, as our model does not support edge features.
> > >
> > > **\[W4\]:** *“Very limited focus only on node classification and lacking experimental results for the claims of a generalist model, limited applicability because of missing tasks from edge- and graph-level tasks..”*
> > >
> > > **Reply:** We acknowledge this as a limitation of our work. The current study is intentionally scoped to heterogeneity in labels and node features to maintain a focused and controlled evaluation. Our model does not yet support edge features, which restricts evaluation to node-level tasks. Extending the framework to edge- and graph-level settings would require non-trivial architectural modifications, and we view this as an important direction for future work.
> > >
> > > We would also like to share an exciting new result. On the request of Reviewer WaZQ, we performed a transfer experiment where we fine-tuned our model on a graph classification task. Please refer to our response to Q1 for Reviewer WaZQ for detailed results. Note that these datasets do not include edge features, which made this experiment feasible within our current framework.
> > >
> > > **\[W6\]:** *“Since the focus does not lie on novel methods here, the hyperparameter tuning and overall discussion of architecture choices remains very limited, thereby not really supporting the claims made by the authors.”*
> > >
> > > **Reply:** We hope we have resolved concerns about our paper's claims through our responses above\! If not, we would be happy to address any further concerns.

---

> > > > ### Comment · Reviewer_dSYo · 2025-10-20
> > > > **Response to author rebuttal**
> > > >
> > > > Dear authors,
> > > >
> > > > thanks a lot for the thoughtful reply and the addition of requested changes.
> > > >
> > > > Regarding your rebuttal I see most of my concerns resolved. I agree that rewording the sections on SOTA performance as proposed would help to avoid misunderstandings concerning the paper. Besides, the other contributions as listed in your reply would be additionally highlighted as compared to a supposed SOTA performance.
> > > > Given the changes in the rebuttal, I agree that the claims are backed by accurate and clear evidence.
> > > >
> > > > One remaining concern is the restriction to node level experiments and no usage of edge features as noted by reviewers SgFC and gPND. Also I would like to see the performance of GraphFM on large scale datasets to get a better estimate of the performance of GraphFM. However, I agree that these experiments are out of the scope of the current paper and may constitute furture work.

---

### Review · Reviewer_WaZQ · 2025-10-07

**Summary Of Contributions:**

This paper presents GraphFM, a Perceiver-based transformer architecture for large-scale multi-graph pretraining across diverse domains. Rather than training a separate GNN for each dataset, the model learns a shared latent space through cross-attention between node embeddings and a fixed set of learnable latent tokens. This design allows the same model to handle graphs of different sizes and topologies efficiently.

To scale training across many datasets, the authors introduce a multi-graph packing scheme that avoids excessive padding, and a Distributed Snake Strategy Sampler that balances GPU utilization when mixing small and large graphs. The model is pretrained on 152 datasets spanning social, citation, and molecular graphs (7.4M nodes, 189M edges), and evaluated on unseen datasets.

Results show consistent improvements with larger model and dataset sizes, and competitive performance with specialized GNNs and graph transformers, all while requiring minimal hyperparameter tuning. The paper also analyzes model stability, showing that pretrained representations lead to smoother optimization and less sensitivity to hyperparameter choice. Overall, the work demonstrates that cross-domain graph pretraining can yield broadly transferable representations.

**Audience:**

Yes

**Audience Explanation:**

This paper aims to determine whether foundation-style pretraining can generalize across graph domains and does so at a scale not previously demonstrated. The work will appeal to researchers in graph learning, transfer learning, and large-scale representation learning. While more of a systems contribution than a theoretical one, the scope and execution make it broadly relevant to TMLR’s audience.

**Broader Impact Concerns:**

This is a methodological paper with no obvious societal risks. The focus is on scalable representation learning and training infrastructure, so a dedicated broader impact statement is not necessary beyond standard comments on compute cost and data provenance.

**Claims And Evidence:**

Yes

**Claims Explanation:**

I believe the claims made are supported by clear and solid empirical evidence.

1. The scaling analysis (Fig. 4) shows a clear monotonic relationship between pretraining scale and downstream accuracy.

2. The efficiency gains from the sampler are substantial (roughly a fivefold speed-up) and well documented.

3. Fine-tuning experiments under both MFT and NFT settings confirm robustness and stable convergence across tasks.

4. Figures 6 and 7 demonstrate markedly lower sensitivity to hyperparameters compared with GCN and NAGphormer.

5. Comparisons with recent graph foundation models, including GraphAny, are given in the appendix and show that GraphFM remains competitive across a range of domains.

**Requested Changes:**

A list of my requested changes, as well as whether they are critical for securing my recommendation for acceptance, is given below:

1. I would like the authors to demonstrate generality beyond node classification by including at least one graph-level or link prediction task. (critical for acceptance)

2. Please quantify the contribution of synthetic datasets through an ablation study comparing training with and without them. I think this is a very interesting an important angle to investigate. (critical for acceptance)

3. Add a brief interpretability or probing analysis to illustrate what information the latent tokens capture. (not critical, but recommended)

4. Include a short discussion of scalability limits, i.e. how runtime and memory scale with latent count K and graph size. (not critical, but recommended)

5. I would also like to suggest some minor corrections below:

- (bottom of page 3): "tthan Ng" -> "than Ng"
- Clearer captions for Figures 4 and 6 (i.e. adding a brief sentence about the overall purpose) would improve paper quality

---

> ### Author Response · Authors · 2025-10-17
> **Response to Reviewer WaZQ**
>
> Thank you for your time and thoughtful review. We are glad that you think our claims are backed by solid empirical evidence. We have incorporated the major requested changes in the revised PDF (highlighted in blue) and provide point-by-point responses below.
>
> **Requested Changes:**
>
> **\[Q1\]:** *“I would like the authors to demonstrate generality beyond node classification by including at least one graph-level or link prediction task. (critical for acceptance)”*
>
> **Reply:** Thank you for the helpful suggestion. To demonstrate the generality of our approach beyond node classification, we conducted additional experiments on graph-level prediction tasks using molecular datasets (MUTAG and PROTEINS). Specifically, we extend GraphFM by adding a cross-attention based decoder after the Perceiver encoder, which aggregates information from the latent tokens to predict the graph label. We apply MLP fine-tuning while keeping the cross-attention decoder learnable. As shown in Table R1, these results demonstrate that GraphFM readily adapts to graph-level prediction, further supporting its generality across diverse task types.
>
> *Table R1: Graph classification results for GraphFM (baselines reported from \[1\])*
>
> |  | MUTAG | PROTEINS |
> | :---- | :---- | :---- |
> | GCN | 74.6 ± 7.7 | 73.1 ± 3.8 |
> | GraphSAGE | 74.9 ± 8.7 | 73.8 ± 3.6 |
> | GT | 75.5 ± 7.9 | 68.4 ± 3.3 |
> | Graphormer | 74.4 ± 7.4 | 68.4 ± 3.7 |
> | SAT | 81.4 ± 8.2 | 71.7 ± 3.5 |
> | SpecFormer | 83.7 ± 8.0 | 72.0 ± 4.9 |
> | **GraphFM** | **93.8 ± 6.6** | **76.4 ± 3.7** |
>
> **\[Q2\] :** *“Please quantify the contribution of synthetic datasets through an ablation study comparing training with and without them. I think this is a very interesting and important angle to investigate. (critical for acceptance)”*
>
> **Reply:** Thank you for this valuable suggestion. This is indeed a really important ablation that will significantly improve the paper. To assess the impact of synthetic datasets, we conducted an ablation study comparing pre-training with and without synthetic graphs. As shown in Table R2, including synthetic datasets improves downstream performance, confirming that they provide useful structural diversity and help the model generalize across graph types.
>
> *Table R2: Mean downstream performance of the largest GraphFM model when pre-trained on all datasets versus excluding synthetic graphs.*
>
>
> |  | Mean Performance |
> | :---- | :---- |
> | All Data | 0.7502 |
> | All data excluding synthetic | 0.7345 |
>
> **\[Q4\] :** *“Include a short discussion of scalability limits, i.e. how runtime and memory scale with latent count K and graph size. (not critical, but recommended)”*
>
> **Reply:** The paper already includes a discussion of the computational complexity of GraphFM with respect to the latent count $K$ and graph size $N\_g$ in section 3.1.1 (page 4). Specifically, the overall complexity scales as $KN\_g \+ LK^2 \\ll N\_g^2$. Please let us know if you meant something else\!
>
> **\[Q5\]:** *“I would also like to suggest some minor corrections below [...]"*
>
> **Reply:**  Thank you for your suggestions. We have corrected the typo and revised the captions for Figures 4 and 6 to include a brief statement of their overall purpose. Please refer to the updated captions on pages 8 and 10 of the revised manuscript.
>
> ---
>
> \[1\] Guan, S., Song, L., Chen, X., Li, Y., Si, Q., Albera, L., ... & Shu, H. Enhancing Graph Tasks with a Dual-Block Graph Transformer: A Synergistic Approach to Local and Global Attention.

---

> > ### Author Response · Authors · 2025-10-31
> > **Reminder Regarding Rebuttal Feedback**
> >
> > Thank you again for your thoughtful and constructive feedback on our manuscript. Your comments have greatly improved the quality of the work. We would like to kindly follow up and ask whether you are satisfied with our rebuttal, or if you would like us to include any additional experiments or revisions.

---

> > > ### Comment · Reviewer_WaZQ · 2025-11-09
> > > **Thank you for the rebuttal**
> > >
> > > Overall, the authors have addressed my critical issues well. The additional experiments strengthen the paper and confirm the main claims. I am satisfied with the revisions and believe this paper meets the threshold for acceptance to TMLR.

---

### Decision · Action_Editor_52Fc · 2025-11-11

**Recommendation:** Accept as is

**Audience:**

Yes

**Audience Explanation:**

The work will appeal to researchers in graph learning, transfer learning, and large-scale representation learning, thus being of interest to the TMLR audience.

**Claims And Evidence:**

Yes

**Claims Explanation:**

The paper introduces GraphFM, a novel pretraining approach which can learn from various graph domains and use the resulting model as a pretrained foundation model to perform node prediction tasks on graphs from unseen domains. The paper proposes techniques to allow scaling up graph training, allowing GraphFM to be trained on over 150 diverse datasets including homophilous and heterophilous graphs. The empirical results support the claim that foundation-style pretraining can generalize across graph domains.

The reviewers requested changes to put the main contributions in perspective (e.g., the main contribution is a new method of data dependent scaling, bit the proposed model does not achieve SOTA results on most datasets) and there is agreement that, after these changes, the main claims in the paper are well supported.